# Co-evolution of matrisome and adaptive adhesion dynamics drives ovarian cancer chemoresistance

Elina A. Pietilä [1], Jordi Gonzalez-Molina[2,3], Lidia Moyano-Galceran [2], Sanaz Jamalzadeh [4], Kaiyang Zhang [4], Laura Lehtinen [5], S. Pauliina Turunen [2], Tomás A. Martins [1], Okan Gultekin [2,3], Tarja Lamminen[5], Katja Kaipio[5], Ulrika Joneborg [6,7], Johanna Hynninen [8], Sakari Hietanen[8], Seija Grénman[8], Rainer Lehtonen [4], Sampsa Hautaniemi [4], Olli Carpén[4,5,9], Joseph W. Carlson [3] & Kaisa Lehti [1,2,10 ✉]

Due to its dynamic nature, the evolution of cancer cell-extracellular matrix (ECM) crosstalk, critically affecting metastasis and treatment resistance, remains elusive. Our results show that platinum-chemotherapy itself enhances resistance by progressively changing the cancer cell-intrinsic adhesion signaling and cell-surrounding ECM. Examining ovarian high-grade serous carcinoma (HGSC) transcriptome and histology, we describe the fibrotic ECM heterogeneity at primary tumors and distinct metastatic sites, prior and after chemotherapy. Using cell models from systematic ECM screen to collagen-based 2D and 3D cultures, we demonstrate that both specific ECM substrates and stiffness increase resistance to platinum-mediated, apoptosis-inducing DNA damage via FAK and β1 integrin-pMLC-YAP signaling. Among such substrates around metastatic HGSCs, COL6 was upregulated by chemotherapy and enhanced the resistance of relapse, but not treatment-naïve, HGSC organoids. These results identify matrix adhesion as an adaptive response, driving HGSC aggressiveness via co-evolving ECM composition and sensing, suggesting stromal and tumor strategies for ECM pathway targeting.

[1] Individualized Drug Therapy Research Program, Research Programs Unit, University of Helsinki, Helsinki, Finland. [2] Department of Microbiology, Tumor, and Cell Biology, Karolinska Institutet, Stockholm, Sweden. [3] Department of Oncology-Pathology, Karolinska Institutet, Stockholm, Sweden. [4] Research Program in Systems Oncology, Research Programs Unit, Faculty of Medicine, University of Helsinki, Helsinki, Finland. [5] Research Center for Cancer, Infections and Immunity, Institute of Biomedicine, University of Turku, Turku, Finland. [6] Department of Women's and Children's Health, Division Obstetrics and Gynaecology, Karolinska Institutet, Stockholm, Sweden. [7] Department of Pelvic Cancer, Theme Cancer, Karolinska University Hospital, Stockholm, Sweden. [8] Department of Obstetrics and Gynecology, Turku University Hospital, University of Turku, Turku, Finland. [9] HUS Diagnostic Center, Helsinki University Hospital, Helsinki, Finland. [10] Department of Biomedical Laboratory Science, Norwegian University of Science and Technology, Trondheim, Norway. ✉email: kaisa.lehti@ntnu.no

Extracellular matrix (ECM)-derived biochemical and biomechanical cues in the tumor microenvironment (TME) critically contribute to cancer cells' ability to metastasize and resist treatment[1]. Together with genetic alterations, ECM controls cancer cell morphology, invasion, survival, and growth[2,3]. During cancer progression, the TME undergoes major changes in both biochemical composition and biomechanical properties, including tissue stiffness[4]. However, the specific effects of metastasis and chemotherapy on the ECM composition and properties have remained incompletely understood.

The ECM is a dynamic and complex molecular network with distinctive biochemical and structural characteristics[5]. Defined as the matrisome, it consists of a group of proteins encoded by genes for core ECM proteins (collagens, proteoglycans, and ECM glycoproteins) and ECM-associated proteins (proteins structurally resembling ECM proteins, ECM remodeling enzymes, and secreted factors)[6]. By providing stem cell niches and regulating cell growth, the ECM controls cell signaling essential to maintain normal tissue functions, whereas changes of ECM structure and mechanics are sufficient to actively promote tumor progression[1,7].

External ECM-mediated stimuli are transduced via adhesion molecules, mainly integrins[8], which, integrated into focal adhesion (FA) complexes, link the cell cytoskeleton to the ECM to reciprocate forces between cells and the microenvironment[9]. Conformational changes in FAs due to such receptor-mediated mechanotransduction lead to cytoskeletal tension and intracellular signaling, which in turn regulate adhesion, migration, and ECM remodeling[10].

High-grade serous carcinoma (HGSC) is the most common form of ovarian cancer (OC) and the deadliest gynecological disease[11]. HGSC patients are most often diagnosed at an advanced stage, with malignant ascites fluid and widely spread metastases accumulated within the peritoneal cavity and organs. The current standard of care is based on surgical cytoreduction and platinum-based chemotherapy, but despite initial good response to the regimen, over 70% of patients develop platinum resistance within 5 years leading to short life expectancy[12].

Disseminated from the ovary or fallopian tube, multicellular HGSC spheroids accumulate in the ascites and spread intraperitoneally within the abdominal cavity lining and into visceral tissues, especially the omentum[13,14]. Metastasis to the omentum results in a transformation of the tissue, primarily composed of adipocytes, to an ECM-rich fibrotic TME, also called desmoplasia, histologically devoid of adipocytes[15]. Specific ECM and collagen-remodeling signatures in HGSC associate with tumor stiffness and extension of the desmoplastic area referred as high disease score, as well as metastasis and poor survival[16,17]. Aside from altering the tumor cells, chemotherapy affects the tumor stroma, including the induction of dense fibrosis and inflammation[18,19]. However, chemotherapy-induced alterations in the matrisome and the relationship between these changes and cellular responses, which could shed light on the processes of HGSC chemoresistance and relapse, have not yet been systematically identified.

To understand how the ECM co-evolves with cancer cell functions in HGSC metastases and chemoresistance, we conducted comprehensive transcriptomics analysis using longitudinal HGSC cohort of distinct anatomical sites pre- and post-chemotherapy. Coupled with functional experiments assessing the effects of ECM and stiffness in cancer cell chemo-responses, we show here that, in the treatment-escaping HGSC cells, platinum induces cell adhesion, spreading, and migration in a manner dependent on both ECM protein composition and stiffness. Our data, including the upregulation of stiffness-dependent tumor-promoting collagen 6 (COL6), describes how metastasis and chemotherapy-induced changes provide unique niches for cancer cells to engage altered ECM remodeling and sensing. This highlights stromal pathways as candidate targets to improve the chemotherapy effectiveness against treatment-escaping HGSC cells.

## Results

**The matrisomes of pre-chemotherapy omental and peritoneal metastases differ from primary tumor matrisome.** To understand how the cancer cell-adjacent ECM microenvironment evolves during HGSC progression and upon treatment, we characterized all known 1027 matrisome genes encoding the core ECM and associated proteins[6] in pre- and post-chemotherapy tumor samples by RNA-sequencing (Fig. 1a; see Supplementary Data 1 and 2 for patient information including treatment arm and survival data).

Pre-chemotherapy, 104 genes were differentially expressed in omental metastases ($n = 21$) and 99 genes in peritoneal metastases ($n = 28$) compared to primary ovary/fallopian tube tumors ($n = 32$), including 50 common differentially expressed genes (DEGs) in both metastatic sites (Fig. 1b). In mesenteric metastases, we found 41 DEGs ($n = 6$; Fig. 1b). Among all three metastatic locations, seven DEGs were shared (COL17A1, COLEC11, INHA, ITIH2, MUC13, PCSK6, TDGF1) and most of all the identified DEGs encoded core matrisome components (Fig. 1b–d, Supplementary Fig. 1a, b, and Supplementary Data 3–6).

While gene expression in the solid tumor tissues can reflect both the cancer and adjacent stromal cells, the HGSC ascites include cancer and immune cells but only a minor representation of the ECM-producing stroma[13]. Compared to the ascites-derived cells ($n = 9$), we identified 384 matrisome DEGs in the solid tumor tissues, largely encoding the core ECM (Fig. 1e and Supplementary Data 7, including all five COL6 encoding genes, several other collagens and FN1). This is consistent with the concept that stromal rather than cancer cells are the main producers of collagens, proteoglycans, and matrix glycoproteins in the solid TME[20].

COL1 and fibronectin 1 (FN1) immunohistochemistry (IHC) of control omentum ($n = 3$) and pre-chemotherapy HGSC ($n = 9$) tissues validated the substantial desmoplastic stromal reaction around tumor cells, where both primary tumor and solid omental metastasis had developed prominent ECM surrounding the malignant cells upon diagnosis (Fig. 1f–h and Supplementary Fig. 1c–f).

To systematically identify pathways enriched by the DEGs in omental and peritoneal metastases, we performed Ingenuity Pathway Analysis (IPA). Based on the matrisome transcriptomes, 70 canonical pathways were enriched in the metastatic omentum and peritoneum tissues compared to primary ovary/fallopian tube tumors. The most significantly upregulated pathway activity was "Inhibition of Matrix Metalloproteases" (MMPs; Fig. 1i and Supplementary Data 8 and 9; $-\log10$($p$ value) = 10.1, $z$-score 0.707; see Supplementary Fig. 1b for corresponding volcano plot). The most significantly altered other pathways included "Granulocyte Adhesion and Diapedesis", "Agranulocyte Adhesion and Diapedesis" (i.e., mononuclear leukocytes), and "Airway Pathology in Chronic Obstructive Pulmonary Disease" (Fig. 1i; $-\log10$ ($p$ value) $\geq 9.91$), highlighting the close association of fibrotic signaling and ECM alterations with immune cells in the metastatic tissues.

**Post-chemotherapy HGSCs exhibit strong core ECM signatures coupled to MMP induction at metastatic sites.** To examine next the chemotherapy-induced transcriptional ECM alterations in both the primary and metastatic tumors, we compared the matrisome gene expression in each tumor site pre- vs post-chemotherapy. In post-chemo primary ovary/fallopian tube

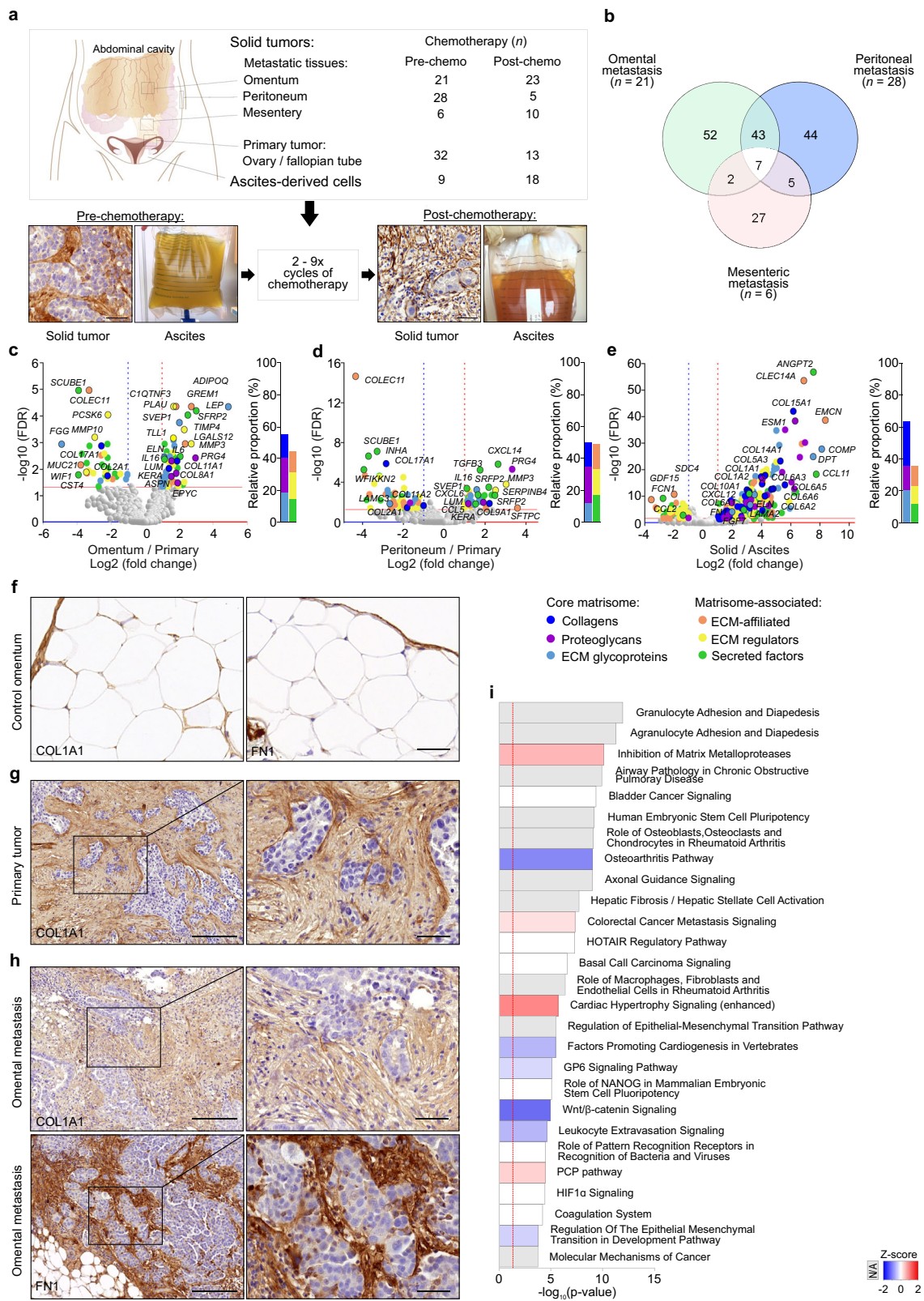

tumors ($n = 13$), we identified 248 matrisome DEGs and in mesenteric metastases ($n = 10$) 266 DEGs, whereas omental metastases ($n = 23$) displayed 182 and peritoneal metastases ($n = 5$) 49 DEGs when compared to the corresponding pre-chemo tumors (Fig. 2a). Collectively, the solid omental, peritoneal, and mesenteric metastases shared 11 post-chemo DEGs (*CHRDL1*, *CLEC3B*, *COL14A1*, *CYR61*, *ECM2*, *EMCN*, *FGF10*, *MMP19*,

*PARM1*, *SEMA3G*, *SNED1*; Supplementary Data 10). The majority of the chemotherapy-altered genes at each tumor location encoded core ECM proteins, among which collagens were proportionally more altered upon chemotherapy in ovary/fallopian tube tumors than in the metastatic sites (Fig. 2b, Supplementary Fig. 2a–c, and Supplementary Data 11–14). Compared to the corresponding pre-chemotherapy tissues, genes for core

**Fig. 1 The matrisomes of pre-chemotherapy omental and peritoneal metastases differ from primary tumor matrisome. a** Schematic diagram, micrographs, and images showing the anatomical locations and the types of samples collected from high-grade serous carcinoma (HGSC) patients, $n$ = number of samples. Scale bar = 20 μm. **b** Venn diagram illustrates the number of differentially expressed genes (DEGs) encoding matrisome proteins in pre-chemotherapy HGSC omental ($n$ = 21), peritoneal ($n$ = 28), and mesenteric ($n$ = 6) metastatic tissues in comparison to primary tumor ($n$ = 32). **c–e** Volcano plots and corresponding charts indicate a strong core matrisome expression in omental (**c** $n$ = 21) and peritoneal (**d** $n$ = 28) metastases against primary tumor ($n$ = 32) and in all solid tissues (primary + metastatic, $n$ = 88) against ascites-derived cells (**e** $n$ = 9). In volcano plots, colored dots indicate DEGs of each matrisome category: dark blue = collagens, purple = proteoglycans, light blue = extracellular matrix (ECM) glycoproteins, orange = ECM-affiliated proteins, yellow = ECM regulators, green = secreted factors. Horizontal line shows Benjamini–Hochberg-adjusted $p$ value <0.05 (FDR false discovery rate); vertical lines depict 2.0-fold increased (red) and decreased (blue) expression. Bars in charts depict the relative proportion of DEGs within the six different matrisome categories. **f–h** Micrographs of collagen 1A1 (COL1A1) and fibronectin (FN1) immunohistochemistry of control omentum (**f**), pre-chemo primary tumor (**g** COL1A1), and omental metastasis (**h**) show substantial desmoplastic reaction surrounding malignant cell areas. Images representative of three patients (**f**) and eight patients (**g, h** see Supplementary Fig. 1c–f for more examples). Scale bar = 0 μm (**f**), 200 μm and 50 μm in inset (**g, h**). **i** Ingenuity Pathway Analysis demonstrates the top pathways affected by DEGs in pre-chemo omental and peritoneal metastases ($n$ = 49) against primary tumor ($n$ = 32); the color key identifies the $z$-score; Fisher's exact test; N/A not applicable.

matrisome including proteoglycans were proportionally highly altered in peritoneal metastases, whereas matrisome-associated factors such as secreted cytokines and chemokines were altered broadly in omental metastasis (Supplementary Fig. 2a, b). At mesenteric metastases, ECM affiliated, ECM regulators as well as secreted factors were broadly altered, mainly enhanced, after chemotherapy (Supplementary Fig. 2c). In the combined analyses of omental and peritoneal metastases, we identified 241 post- vs pre-chemo matrisome DEGs, consisting almost equally of core matrisome and matrisome-associated genes (Fig. 2c and Supplementary Data 15). Only 24 chemotherapy-altered matrisome DEGs, mostly encoding secreted factors and glycoproteins, were instead found in ascites-derived cells ($n$ = 18; Fig. 2d and Supplementary Data 16).

Despite the prominent chemo-induced core ECM signature in primary ovary/fallopian tube tumors, the post-chemo omental and peritoneal metastases retained stronger expression of fibrillar (COL1, COL3, COL5, COL11) and microfibrillar (COL6) collagens compared to the post-chemo ovary/fallopian tube tumors (Fig. 2e and Supplementary Data 17). Collectively, the 267 identified DEGs in the post-chemo omental and peritoneal metastases compared to post-chemo ovary/fallopian tube tumors correlated with altered activity of "Granulocyte Adhesion and Diapedesis" and "Agranulocyte Adhesion and Diapedesis", which were followed by "Airway Pathology in Chronic Obstructive Pulmonary Disease", "Hepatic Fibrosis/Hepatic Stellate Cell Activation", and "Axonal Guidance Signaling" quite similarly with the pre-chemo state (Fig. 2f and Supplementary Data 18; −log10($p$ value) ≥ 23.4). Opposite to the comparison between pre-chemo primary and metastatic tumors, however, the pathway activity for "Inhibition of MMPs" was reduced upon higher MMP expression in post-chemo omental and peritoneal metastases compared to the post-chemo primary tumors (Fig. 2f; −log10($p$ value) = 22.8, $z$-score −2.4).

To further compare matrisome expression in chemo-sensitive vs resistant disease, both pre- and post-chemotherapy, we segregated the patients into the groups of chemo-resistant (platinum-free interval, PFI ≤ 6 months) or chemo-sensitive (PFI > 6 months) disease (Supplementary Fig. 3a and Supplementary Data 19–22). This comparison revealed 22 DEGs in ovary/fallopian tube tumors pre-chemo, encoding mainly ECM glycoproteins, ECM-affiliated proteins, and secreted factors (Supplementary Fig. 3b; resistant $n$ = 6, sensitive $n$ = 23), whereas post-chemotherapy the total of 32 DEGs included collagens (COL11A1, COL10A1, and COL3A1) and chemokines, mostly upregulated, in patients with resistant disease (Supplementary Fig. 3c; resistant $n$ = 5, sensitive $n$ = 8). In solid omental, peritoneal plus mesenteric metastases pre-chemo, only 11 out of 73 DEGs were instead overexpressed in patients with

resistant disease, whereas various secreted and core matrisome genes including IL6 and COL11A2 were lower than in the metastasis of sensitive disease (Supplementary Fig. 3d; resistant $n$ = 24, sensitive $n$ = 27). Post-chemo, CCL28, MUC4, BGN, and S100A11 were overexpressed in the metastatic tissues of resistant patients, whereas the 11 DEGs consisting of non-collagen genes were downregulated (Supplementary Fig. 3e; resistant $n$ = 12, sensitive $n$ = 23; see Supplementary Data 23 for shared DEGs and Supplementary Data 24 and 25 for ascites-derived cancer cells).

By IHC, even the micro-metastatic cancer colonies detected in routinely operated, visually unaffected HGSC post-chemotherapy omentectomy samples ($n$ = 2) were tightly surrounded by accumulating COL1 and FN-rich ECM (Fig. 2g, h and Supplementary Fig. 4a, b), highlighting the development of fibrotic TME even in micro-metastasis. However, in the post-chemo mesenteric and omental HGSC metastases ($n$ = 12), the ECM appeared less dense and more fragmented than in the corresponding pre-chemo tissues. These compromised stromal ECM fibers were surrounded by small cells similar to those detected as CD45[+] immune cells (Fig. 2i, j and Supplementary Figs. 4c–f and 5a, b).

Altogether these data demonstrate that the fibro-inflammatory TME, closely surrounding the cancer foci, is markedly different in primary vs metastatic tumors and changes in response to chemotherapy, including alterations on both the extent and type of the cancer-adjacent core ECM.

**ECM signaling is altered in HGSC cells by matrix stiffness and platinum treatment.** The evolving tumor matrisome can alter cancer cell functions at least by (1) biomechanical signaling depending on the extent of collagenous ECM and consequent tumor stiffness and (2) adhesion signaling depending on the type of ECM substrates. To examine first the signaling response of HGSC cells to in vivo-like, physiological low and high stiffness range reported in HGSC omental metastasis tissues (0.40–33.13 kPa)[16], we used soft (2 kPa) and stiff (21 kPa) poly-acrylamide hydrogels functionalized for cell adhesion with covalently bound COL1. As relevant cell models for the heterogeneous HGSC cell phenotypes, we used relatively platinum-sensitive, epithelial (CDH1[+], CDH2[low]) OVCAR4, more resistant and mesenchymal (CDH1[−], CDH2[+]) OVCAR8, and the platinum-sensitive, mesenchymal (CDH1[−], CDH2[low]) TYK-nu, all of HGSC origin and harboring TP53 mutations[19,21–23] (see Supplementary Fig. 6a for corresponding platinum sensitivities in our standard two-dimensional (2D) culture).

In response to increasing stiffness, spreading of all these cells enhanced significantly (Fig. 3a, b; $p \leq 2.2 \times 10^{-7}$) in conjunction with FA formation, detectable exclusively on the stiff matrix by

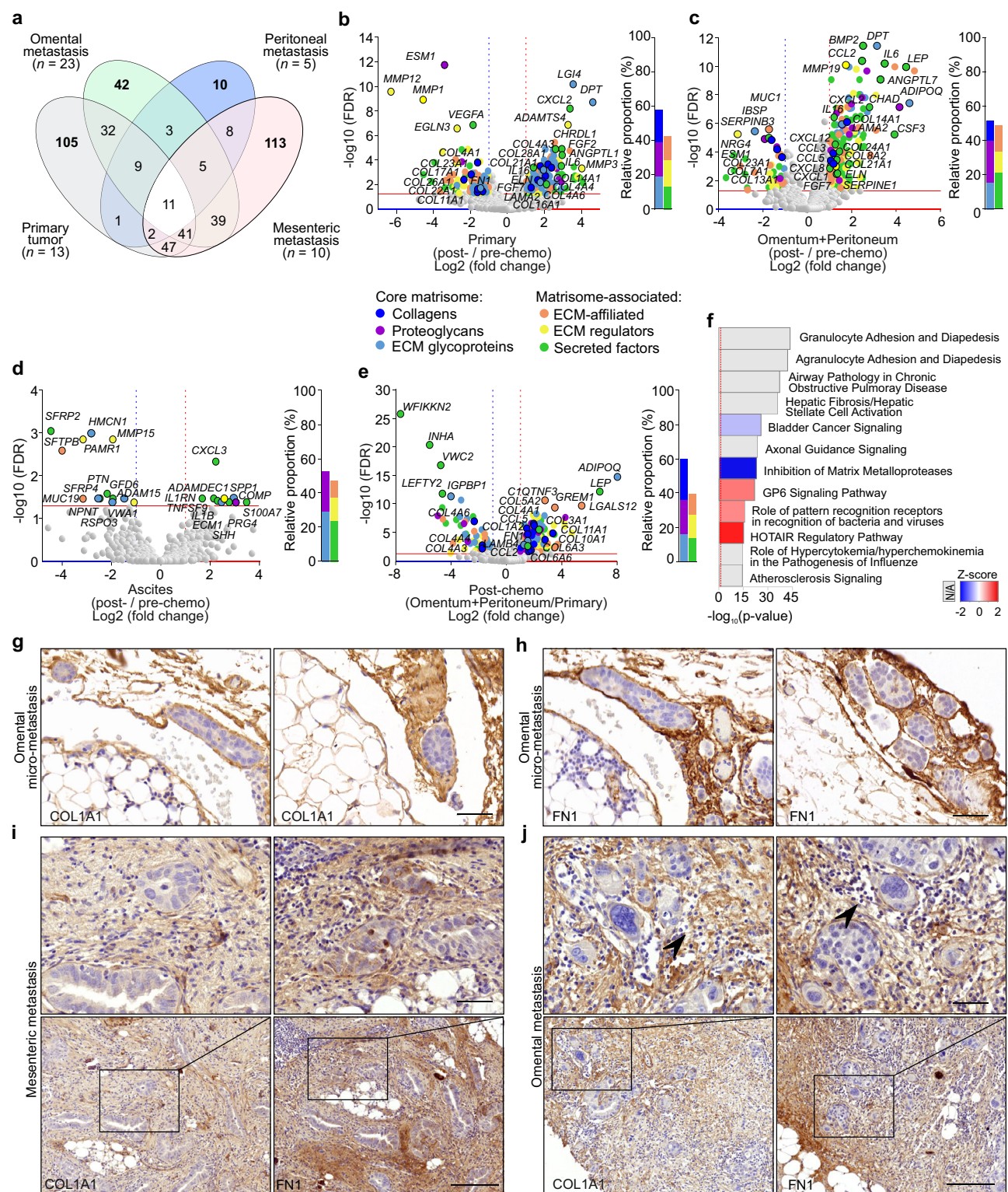

phosphorylated FA kinase (pFAK; Fig. 3a). Coincidentally, nuclear localization of the mediators of ECM mechanical signaling YAP/TAZ increased (Fig. 3c, d; $p < 0.0001$), indicating that stiffness alters ECM cell adhesion and signaling, as expected[24]. Of note, cisplatin treatment also enhanced cell spreading (Supplementary Fig. 6b; $p < 0.001$) as well as peripheral pFAK$^+$ FAs (Supplementary Fig. 6b; $p < 0.006$) in OVCAR4 and OVCAR8 on extremely stiff glass surface. In conjunction, F-actin anisotropy was reduced (Supplementary Fig. 6c; $p \leq 0.024$),

resulting in loss of the transversal fibers and formation of branched radial fibers. These results suggest that the same ECM–FA signaling axis controlled by matrix stiffness is also altered upon HGSC cell response to platinum treatment.

**Increased ECM stiffness protects HGSC cells against cisplatin-induced apoptosis via FAK and YAP signaling.** To investigate whether the biomechanical signaling at increasing matrix stiffness functionally contributes to platinum resistance of HGSC cells,

**Fig. 2 Post-chemotherapy HGSCs exhibit strong core ECM signatures coupled to MMP induction at metastatic sites. a** Venn diagram illustrates the number of differentially expressed genes (DEGs) encoding matrisome proteins in post-chemotherapy HGSC tissues against corresponding pre-chemotherapy tissues. The number of samples post-chemo vs pre-chemo was 23 vs 21 for omental metastasis, 5 vs 28 for peritoneal metastasis, 10 vs 6 for mesenteric metastasis, and 13 vs 32 for primary tumor. **b–e** Volcano plots and corresponding charts indicate the strong core matrisome expression in post-chemotherapy primary (ovary/fallopian tube) tumors (**b** $n = 13$ vs 32), combined omental and peritoneal metastases (**c** $n = 28$ vs 49), and in ascites-derived cells (**d** $n = 18$ vs 9) compared to matching pre-chemotherapy samples. Post-chemotherapy, DEGs between combined omental and peritoneal metastases against the primary tumor are shown in **e** ($n = 28$ vs 13). In volcano plots, colored dots indicate DEGs of each matrisome category: dark blue = collagens, purple = proteoglycans, light blue = extracellular matrix (ECM) glycoproteins, orange = ECM-affiliated proteins, yellow = ECM regulators, green = secreted factors. Horizontal line shows Benjamini–Hochberg-adjusted $p$ value <0.05 (FDR false discovery rate); vertical lines depict 2.0-fold increased (red) and decreased (blue) expression. Bars in charts depict the relative proportion of the DEGs within the six different matrisome categories. **f** Ingenuity Pathway Analysis demonstrates the top pathways affected by DEGs in post-chemotherapy omental and peritoneal metastases ($n = 28$) against post-chemotherapy primary tumor tissue (ovary/fallopian tube, $n = 13$); the color key identifies the $z$-score; Fisher's exact test; N/A not applicable. **g–j** Micrographs of collagen 1A1 (COL1A1) and fibronectin (FN1) immunohistochemistry of post-chemotherapy omental micro-metastasis (**g**, **h**) show the development of fibrosis in early micrometastatic tumor microenvironment (TME) and the compromised ECM fibers in post-chemotherapy mesenteric (**i**) and omental (**j**) metastases. Images representative of two patients (**g–i**) and ten patients (**j**). Arrowheads indicate fragmented ECM fibers. See Supplementary Fig. 4a, b for lower magnification micrographs and Supplementary Fig. 4c–f for more examples. Scale bar = 50 µm.

OVCAR4 and OVCAR8 were first seeded on 2, 4.5, and 21 kPa hydrogels followed by cisplatin treatment for 32 h. In both these cells, cisplatin-induced apoptosis (cleaved Caspase3/cl-Casp3) was lower at enhanced stiffness (Supplementary Fig. 7a; OVCAR4: $63.0 \pm 9.1\%$, OVCAR8: $69.1 \pm 15.3\%$ higher cl-Casp3+ cells on 2 vs 21 kPa, $p \leq 0.032$). In contrast, both the proliferation of the untreated cells (5-ethynyl-2′-deoxyuridine, EdU, incorporation) and the cisplatin-induced effective DNA damage (intensity of phosphorylated histone H2Ax, γH2Ax, per nuclei) were higher in OVCAR4 and similar in OVCAR8 in the stiff compared to soft substrate (Supplementary Fig. 7b, c), suggesting that, coincident with increased FAK and YAP signaling, matrix stiffness can increase HGSC resistance to platinum-induced apoptosis.

To further investigate how the stiffness can affect platinum resistance, OVCAR4, OVCAR8, and TYK-nu cells as well as the more resistant TYK-nu.R subline, generated by repeated cisplatin exposure and thus serving as a model for post-chemo/relapse TYK-nu cells[25], were used as cell models. In each cell line and on both soft and stiff matrix, DNA damage accumulated over 36-h cisplatin treatment, as assessed by γH2Ax (Fig. 3e; $p \leq 0.035$; Supplementary Fig. 7d). The total and apoptotic (cleaved Caspase 3/7+/cl-Casp3/7+) cells were measured on soft and stiff matrices by live cell imaging over 72-h cisplatin treatment. Markedly, the increase in apoptotic OVCAR4, OVCAR8, and TYK-nu.R was significantly higher on soft compared to stiff substrate, whereas TYK-nu apoptosis remained unaffected by the stiffness (Fig. 3f and Supplementary Movies 1–4; OVCAR4: $49.6 \pm 11.9\%$, OVCAR8: $88.6 \pm 5.7\%$, TYK-nu.R $52.2 \pm 13.6\%$ higher cl-Casp3/7+ cells on 2 vs 21 kPa at 36 h, $p \leq 0.033$). Moreover, total number of TYK-nu cells declined during the treatment, whereas the low apoptosis rates of OVCAR4, OVCAR8, and TYK-nu.R were coupled with increasing total cell counts on the stiff substrates (Supplementary Fig. 8a). On both soft and stiff substrates, >66% of cyclinA2+ OVCAR4, TYK-nu, and TYK-nu.R cells displayed RAD51+ nuclei, indicative of homologous recombination (HR) proficiency (Fig. 3g and Supplementary Fig. 8b–d)[26]. Despite high cyclinA2 positivity, only <10% of OVCAR8 cells were instead RAD51+, consistent with their reported HR deficiency, generally linked to platinum responsiveness (Fig. 3g and Supplementary Fig. 8b–d)[26]. Therefore, stiffness enhances the resistance of the HR-proficient OVCAR4 and TYK-nu.R as well as the HR-deficient OVCAR8 to apoptosis-inducing DNA damage.

To assess the dependence of the stiffness-mediated cisplatin resistance on adhesion signaling, FAK was inhibited with defactinib in OVCAR4 and OVCAR8 grown on the stiff COL1

hydrogels. In combination with cisplatin, defactinib enhanced apoptosis (Fig. 3h and Supplementary Fig. 8e), although γH2Ax intensity was decreased, especially in OVCAR4 (Fig. 3i). Similarly, inhibition of YAP with verteporfin resulted in enhanced cisplatin-induced apoptosis, while the net DNA damage was lower or equal to cells treated with cisplatin only (Fig. 3j, k and Supplementary Fig. 8f).

Altogether, these results indicate that the stiffness-induced FAK-YAP signaling increases resistance to apoptosis upon platinum-mediated DNA damage.

**Specific matrix proteins alter platinum response of HGSC cells.** To next determine whether the adhesion signaling via specific ECM components in the TME, besides the biomechanical, stiffness-dependent stimuli, mediates platinum resistance, we systematically screened for the effects of different ECM single or combination substrates in OVCAR4/OVCAR8 and TYK-nu/TYK-nu.R cell pairs.

For the single substrates, the adherent untreated OVCAR4 and OVCAR8 cell numbers were highest on FN and COL6 (the highest OVCAR8 count on FN was set to 100.0) and low on elastin (ELN), COL4, and COL3 (Fig. 4a, b and Supplementary Fig. 9a, b). Distinct ECM proteins differentially affected OVCAR4 and OVCAR8 cisplatin response (defined by the change in cell count upon treatment). While treatment resistance of OVCAR4 varied dramatically in an ECM component-dependent manner, platinum response being strongly positive on COL1 and negative on FN and vitronectin (VTN), OVCAR8 counts on COL1 and other collagen-containing combination substrates increased upon treatment, resulting in negative platinum response (Fig. 4c and Supplementary Fig. 10a, b; see negative cisplatin responses indicative of strongest chemoresistance in red).

Similarly, for the untreated TYK-nu, COL6 and FN together with VTN were the most favorable single ECM protein substrates, while TYK-nu.R was most abundant on COL6 combinations (Supplementary Fig. 10c, d), determined by the ATP content as a measure of metabolically active, viable cells. Both cells were most abundant on COL6 + ELN (TYK-nu content on COL6 + ELN was set to 100.0) and least abundant on ELN (Supplementary Fig. 10c, d). Whereas TYK-nu viability was decreased by cisplatin largely in a substrate-independent manner, TYK-nu.R remained viable upon the treatment on all matrices except COL4 + ELN (Fig. 4d and Supplementary Fig. 10e, f).

Of note, cisplatin sensitivities and pre-treatment counts of OVCAR4, OVCAR8, and TYK-nu.R showed similar tendencies (Supplementary Fig. 10g; $r \leq 0.35$). Moreover, the fraction of pre-treatment Ki67+ OVCAR8 seemed to correlate with cisplatin

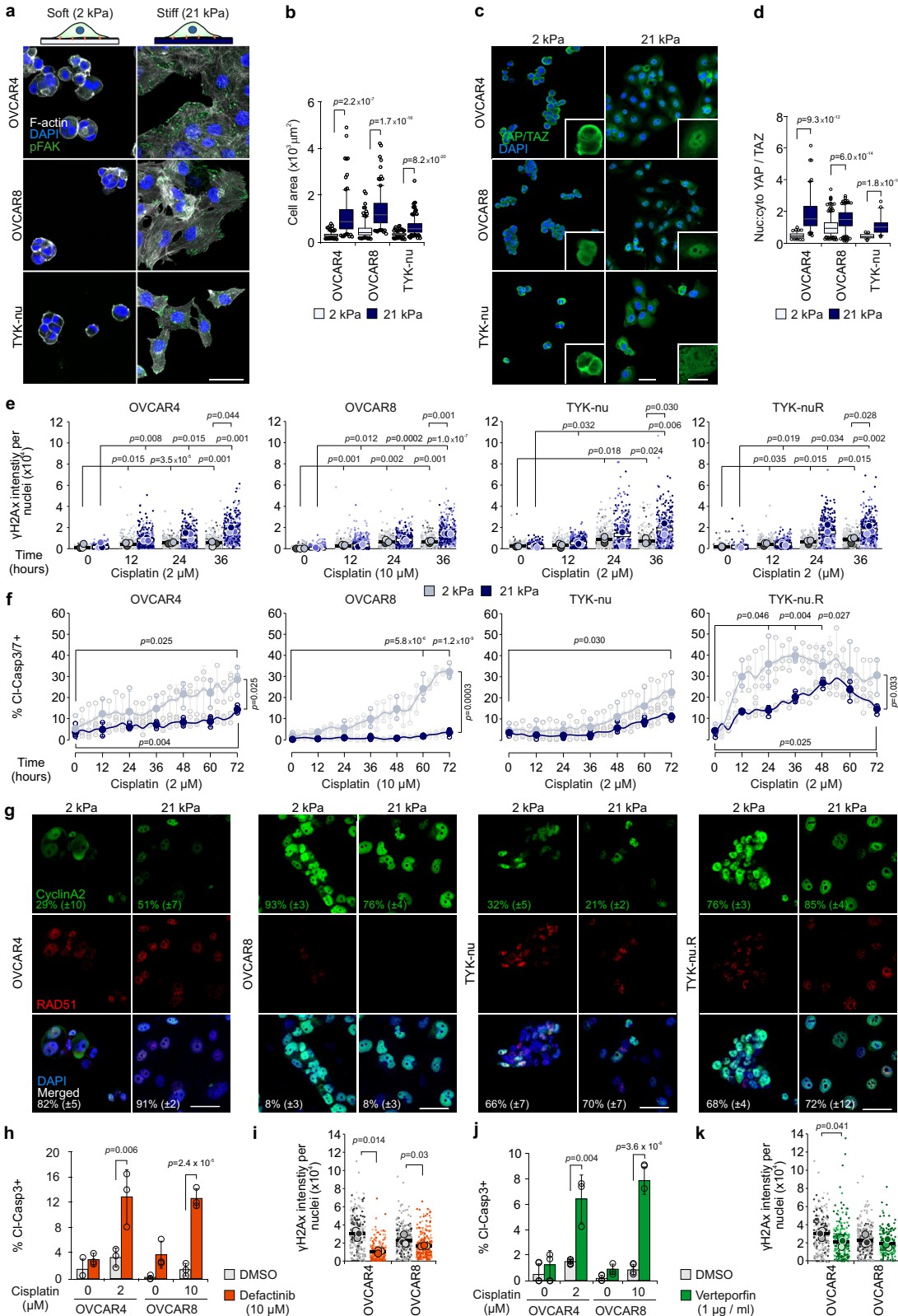

response (Supplementary Fig. 10h; $r = 0.42$), suggesting that at equal stiffness the protective effect of low adherence substrates coincided with reduced proliferation.

To further test this possible link between ECM-mediated cisplatin response and cell growth, we plated in low (1%) serum OVCAR4, OVCAR8, TYK-nu, and TYK-nu.R as well as OVCAR3, a platinum-sensitive, epithelial ($CDH1^+$, $CDH2^{low}$)

cell model[19] on COL6 and FN, i.e., the single ECM substrates globally favorable for HGSC adherence, as well as on VTN and COL1. While the decrease of serum had relatively subtle ECM component-dependent effects on OVCAR3 and OVCAR4 growth, as assessed by change in cell number, the growth of OVCAR8, TYK-nu, and TYK-nu.R reduced significantly regardless of the ECM type (Supplementary Fig. 10i; $0.56 \pm 0.03$-fold

**Fig. 3 Increased ECM stiffness protects HGSC cells against cisplatin treatment. a, b** Schematic diagram of the experimental design and representative confocal micrographs of F-actin (phalloidin, white) and phosphorylated focal adhesion kinase (pFAK, green) in cells on 2 and 21 kPa collagen 1-functionalized polyacrylamide hydrogels (COL1-PAA); OVCAR4 and OVCAR8 $n = 100$, TYK-nu $n = 150$ cells. Scale bar = 25 µm. **c, d** Representative confocal micrographs and corresponding nuclear to cytoplasmic ratio (nuc:cyto) of YAP/TAZ (green) in representative cells on 2 and 21 kPa COL1-PAA; OVCAR4 $n = 101/101$, OVCAR8 $n = 323/300$ and TYK-nu $n = 56/45$ cells. Scale bar = 25 µm and 10 µm in inset. **e** Superplots show the phosphorylated H2Ax (γH2Ax) intensity per nuclei over 36 h on 2 (gray) and 21 kPa (blue) COL1-PAA. Superplots depict each cell within color-coded replicate and their mean. See Supplementary Fig. 7d for representative confocal micrographs. **f** Charts depict the cleaved caspase 3/7+ (cl-casp3/7+) cells over 72 h treatment on 2 and 21 kPa COL1-PAA. See Supplementary Movies 1–4. **g** Representative confocal images and corresponding quantification of cyclinA2 (green), RAD51 (red), and co-expression (merged) in nucleus (DAPI, blue) in corresponding cells on 2 and 21 kPa COL1-PAA at 36 h cisplatin treatment (2 µM OVCAR4, TYK-nu, TYK-nu.R; 10 µM OVCAR8). Standard deviation shown in brackets. See Supplementary Fig. 8b for 24 h treatment and Supplementary Fig. 8c, d for complete quantifications. Scale bar = 50 µm. **h–k** Charts and superplots illustrate cl-Casp3+ cells and γH2Ax intensity per nuclei in cells on 21 kPa COL1-PAA with DMSO (control; gray), Defactinib (**h, i** orange) or Verteporfin (**j, k** green) at 32 h; see Supplementary Fig. 8e, f for representative micrographs of cl-Casp3. Superplots depict each cell within color-coded replicate and their mean. Data represent mean ± SEM; $n = 3$ biological replicates; two-tailed Student's $t$ test; one-way ANOVA with Tukey's multiple comparison test (**f** 0–72 h comparisons within 2 and 21 kPa). Box plots indicate median (middle line), 25th, 75th percentile (box), and 10th and 90th percentile (whiskers) as well as outliers (single points). Source data are provided as a Source data file.

decrease in OVCAR8; $0.35 \pm 0.06$-fold in TYK-nu and 0.39-fold in TYK-nu.R compared to 10% serum; $p \leq 0.002$). While the cisplatin response, determined by the treatment-induced change in the number of OVCAR3, OVCAR4, and TYK-nu likewise varied in an ECM-dependent manner, this did not fully correlate with the corresponding ECM-dependent growth rates (Supplementary Fig. 10j). The effects of the distinct ECM components to cisplatin response of OVCAR8 and TYK-nu.R were instead diminished coincident with suppressed growth in low serum (Supplementary Fig. 10j).

Therefore, the ECM-dependent growth rate can be a central element, but likely not solely responsible, of the prominent and variable ECM component-dependent changes in cellular cisplatin response in HGSC cells.

**VTN, FN, and COL6 activate distinct platinum-induced cell spreading and migratory responses**. To investigate the mechanisms behind the ECM component-dependent platinum responses, we assessed the adhesion-dependent dynamics of OVCAR4 and OVCAR8 on the ECM arrays, focusing on cell spreading, associated with stiffness-induced resistance, and migration. Cisplatin increased OVCAR4 and OVCAR8 spreading area on 24 and 11 ECM substrates, respectively, whereas OVCAR8 spreading was reduced on COL1 + COL6 (Supplementary Fig. 11a, b). The area of pre-treatment OVCAR4 seemed to negatively correlate with that of the post-treatment cells on the same matrices, suggesting that cisplatin can alter OVCAR4 adhesive area least on the substrates supporting constitutively prominent spreading (Supplementary Fig. 11c; $r = -0.46$). However, post-treatment area showed a similar trend with cell count (Fig. 4e–g; OVCAR4: $r = 0.45$; OVCAR8: $r = 0.37$), and OVCAR8 spreading increased along with chemoresistance (Fig. 4h; $r = -0.46$). Among the ECM substrates supporting treatment-induced spreading, VTN, FN, and COL6 were the top single ECM substrates for OVCAR4 and COL1 for OVCAR8 (Fig. 4e–g). Therefore, the ECM component-dependent constitutive and cisplatin-induced spreading were associated with poor platinum response of these cells.

Markedly, OVCAR8 further displayed a broad ECM-dependent migratory response upon treatment, as quantified by the number of cells migrating from the original micro-spot (Fig. 4f and Supplementary Fig. 9b). This platinum-induced migration was strong on VTN, FN, and COL6 alone and in combination with VTN (Supplementary Fig. 11d), and the migration correlated with the pre- and post-chemo cell count (Fig. 4i; $r = 0.58$ and $r = 0.71$, respectively). Based on these results, we conclude that cisplatin alters ECM–cell communication, especially on VTN, FN, and COL6, supporting adhesion, migration, and chemoresistance differentially for the platinum-resistant OVCAR8 and more sensitive OVCAR4 cells.

**COL6 increases upon chemotherapy and is associated with poor patient survival**. To validate the clinical relevance of our findings, we first used The Cancer Genome Atlas (TCGA) transcriptomics microarray data of 538 chemo-naïve OC samples to assess survival associations of COL6, FN1, and VTN expression. In pre-chemotherapy TCGA tumors, predominantly consisting of primary tumors[27], high COL6A2, COL6A3, and FN1 expression was associated with poor overall survival ($p \leq 0.044$), whereas COL6A1, COL6A5, or COL6A6 (encoding the alternative polypeptides to COL6A3 in the trimeric protein[28]) and VTN did not show significant associations (Fig. 5a and Supplementary Fig. 12a).

We next examined the expression of these genes in primary tumor and metastatic tissues in our 167 longitudinally collected HGSC sample dataset, pre- and post-chemotherapy. Expression of COL6A1-A3 and FN1 was high both in primary tumor and metastatic tissues, pre- and post-chemotherapy, while COL6A5-A6 and VTN expression was lower (Fig. 5b; primary tumor pre-chemo $n = 32$/post-chemo $n = 13$; metastatic tissues $n = 55/n = 38$; Supplementary Data 26). In metastatic tissues compared to the primary tumors, the expression of COL6A5-A6 and FN1 was higher both pre- and post-chemotherapy, while COL6A3 was higher post-chemotherapy (Fig. 5b; $p \leq 0.049$; Supplementary Data 27–28).

To understand the clinical meaning of these results per individual patient, we further assessed the correlation of COL6, FN1, or VTN with treatment response in 8 patient- and tumor site-matched metastatic samples. Notably, the increase of COL6A1 and COL6A2 post- vs pre-chemo correlated with shorter PFI ($r \leq -0.79$, $p \leq 0.020$) and progression-free survival (PFS; $r \leq -0.80$, $p \leq 0.017$; Fig. 5c and Supplementary Data 29; see Supplementary Fig. 12b for COL6A3-A6, FN1, and VTN; $n = 8$). Moreover, in patients who initially responded to chemotherapy treatment, but later developed platinum resistance, COL6A1-6A3 increased both in solid tumors and in ascites-derived cells upon chemotherapy, whereas FN1 even decreased and VTN remained low (Fig. 5d; COL6A2 in metastatic tissues $p = 0.049$; see Supplementary Fig. 12c for COL6A5 and COL6A6; $n = 12$). In the ascites-derived cancer cells, COL6A1-6A3 expression was relatively low (Fig. 5d; $n = 4$).

Based on a publicly available dataset of 32 epithelial HGSC and 31 stromal samples[29], the expression of each of the COL6 genes was markedly higher in the stroma than in the epithelial

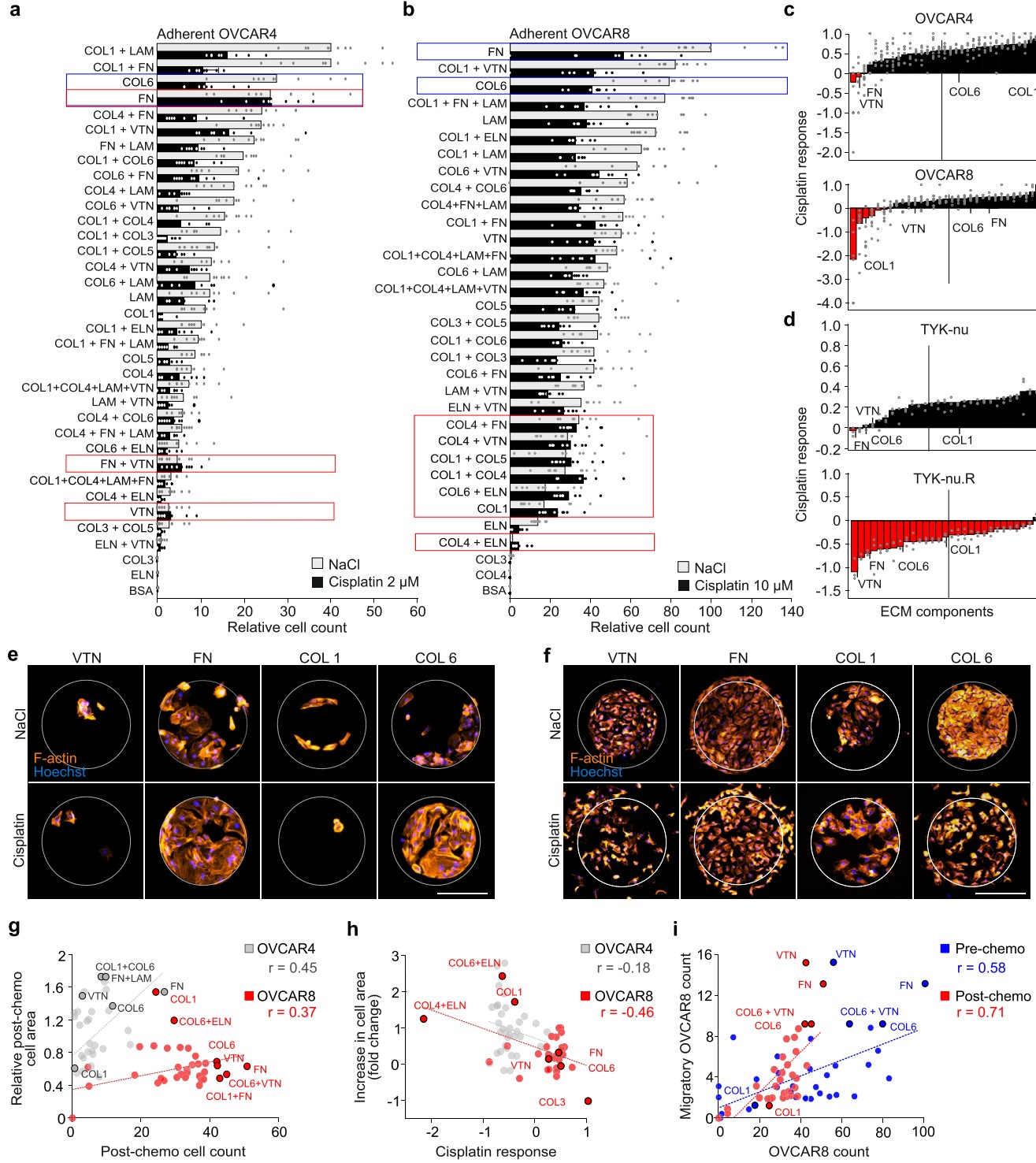

compartment (Fig. 5e). Consistent with this transcriptomics data, COL6 protein was abundant in HGSC omental metastasis-derived cancer-associated fibroblasts (CAFs; $n = 3$) as well as in normal human lung fibroblasts (NFs), whereas COL6 protein was low or undetectable in HGSC cell lines and patient-derived HGSC cells (Fig. 5f; $n = 3$). Furthermore, in omental micro-metastasis ($n = 2$) and desmoplastic, metastatic HGSC tissues ($n = 16$), COL6 protein expression was abundant, with strong expression particularly in the immediate surroundings of the micrometastatic foci as well as in the stroma of early and established metastatic tumors (Fig. 5g).

**Addition of COL6 promotes HGSC cell survival in 3D collagen 1.** Based on our above findings that COL6 and FN, both abundant in the tumor ECM, can differentially support HGSC cell spreading, migration, and chemoresistance, we next investigated their effects in a relevant three-dimensional (3D) model recapitulating the dimensionality of in vivo tumors. To this end, we embedded OVCAR4 and OVCAR8 within two frequently used matrices, the fibrotic tumor stroma-typifying cross-linked COL1 and laminin-rich Matrigel for 5 days. While OVCAR4 grew spherically in both matrices, OVCAR8 grew invasively in COL1, while growing spherically in Matrigel (Fig. 6a).

**Fig. 4 Specific matrix proteins alter HGSC cell platinum response. a–c** Charts depict the OVCAR4 (**a**) and OVCAR8 (**b**) adhesive cell count, and cisplatin response (**c**) on extracellular matrix (ECM) protein array after 24 h adherence and 48 h treatment with 2 μM (OVCAR4) or 10 μM (OVCAR8) cisplatin (see Supplementary Fig. 9a, b for light micrographs). Data in **a**, **b** are presented relative to untreated OVCAR8 cell count in fibronectin (FN). Data are shown as mean ± SEM; $n = 1$ array/biological replicate with 9 technical replicates (**a**, **b**); vertical bars (**c**) indicate mean value. Red boxes (**a**, **b**) and red bars (**c**) indicate the conditions that result in higher cell number after cisplatin treatment compared to NaCl (control). Blue boxes indicate single ECM proteins with highest cell count. **d** Charts depict cisplatin response of TYK-nu and TYK-nu.R on ECM protein array after 24 h adherence and 48 h treatment with 2 μM cisplatin (see Supplementary Fig. 10c, d for quantification). Vertical bars indicate mean value; $n = 3$ biological replicates. Red bars indicate the conditions that result in higher cell number after cisplatin treatment compared to NaCl (control). **e, f** Representative light micrographs of F-actin (phalloidin, orange) in OVCAR4 (**e**) and OVCAR8 (**f**) on ECM micro-spots of vitronectin (VTN), FN, and collagen 1 and 6 (COL1, COL6); $n = 1$ array/ biological replicate, micrographs are representative of 9 technical replicates; see supplementary Fig. 9a, b for full set of micrographs (OVCAR4 $n = 1$ array and OVCAR8 control $n = 2$ arrays, cisplatin $n = 1$ array). Scale bar = 50 μm. **g–i** Scatter plots depict the correlation of post-chemo cell area against cell count (**g**) and the increase in cell area against cisplatin response (**h**) in OVCAR4 and OVCAR8, as well as the number of migrated OVCAR8 against cell count pre- and post-chemotherapy (**i**). Cisplatin response determined by [cell viability (NaCl) − cell viability (cisplatin)]/cell viability (NaCl). Two-tailed Pearson correlation; $n = 1$ array with 9 technical replicates; ELN elastin, LAM laminin, BSA bovine serum albumin. Source data are provided as a Source data file.

Further considering that our matrisome analysis and previous reports define the metastatic HGSC TME as collagenous stroma[30,31], we supplemented the invasion-supportive 3D COL1 matrix with COL6 and FN to understand their pathophysiological relevance to HGSC cell functions (Fig. 6b; see Supplementary Fig. 13a for model validation). As assessed by Ki67, proliferation of both OVCAR4 and OVCAR8 was comparable with and without COL6 or FN (Fig. 6c and Supplementary Fig. 13b, c). However, total ATP measurement showed an increase of cell activity in OVCAR4 and a decrease in OVCAR8 by COL6 (Fig. 6d, e). Notably, upon cisplatin treatment, OVCAR8 viability was higher in COL1 + COL6 compared to COL1 or COL1 + FN (Fig. 6e; increase to COL1 $2.3 ± 0.4$-fold, $p = 0.01$ and to COL1 + FN $1.9 ± 0.3$-fold, $p = 0.031$, with 30 μM cisplatin), whereas OVCAR4 viability remained comparable between the matrices (Fig. 6d). During the 5-day culture, FN1 and to a lesser extent COL6 were incorporated into the 3D ECM around OVCAR8 even without supplementation (Supplementary Fig. 13d–f), indicating that these cells also deposited the culture medium- and/or cell-derived FN into their surrounding COL1 matrix.

Platinum-sensitive OVCAR3 displayed an increase in total ATP when COL6 was added to the 3D COL1 matrix (Fig. 6f), whereas cellular ATP remained unaltered in sensitive TYK-nu and was decreased in resistant TYK-nu.R in the COL6-containing matrix (Fig. 6g, h). Upon cisplatin treatment, only the intrinsically cisplatin-resistant TYK-nu.R had higher ATP-based cell viability in COL1 + COL6 compared to COL1 (Fig. 6f–h). Altogether these results indicate that in the invasively growing OVCAR8 and TYK-nu.R COL6 supports chemoresistance in COL1-based 3D microenvironment.

**Cisplatin treatment enhances integrin-based cancer cell adhesion on stiff COL6 substrate.** To clarify whether the selective cisplatin-protective effect of COL6 is caused by a differential COL6 adhesion upon platinum treatment, and to examine the potential stiffness dependency of this effect, we seeded OVCAR4 and OVCAR8 onto 2, 4.5, or 21 kPa COL6 functionalized gels. Like on COL1, the spreading of OVCAR4 and OVCAR8 enhanced coincident with the increased stiffness (Supplementary Fig. 14a, b), suggestive of improved adhesion. However, YAP/ TAZ nuclear translocation (nuc:cyto ratio >1) occurred at a lower stiffness in OVCAR8 than in OVCAR4 in conjunction with enhanced OVCAR8 proliferation (Fig. 7a–c and Supplementary Fig. 14c; OVCAR8: $1.6 ± 0.1$-fold higher Edu+ cell count on 21 vs 2 kPa, $p = 0.0361$). Cisplatin further enhanced OVCAR4 and reduced OVCAR8 spreading on stiff COL6 in conjunction with diminished cisplatin-induced apoptosis particularly in OVCAR8 at increasing stiffness (Fig. 7d and Supplementary Fig. 14a, b, d;

OVCAR8: $82.0 ± 1.1\%$ higher cl-Casp3+ cell count on 2 vs 21 kPa, $p = 0.0005$).

Considering this COL6-adhesion-dependent survival particularly in the invasive OVCAR8, we analyzed the effects of cisplatin on adhesive properties of the cells on stiff 21 kPa COL1 and COL6. Notably, cisplatin endorsed FA formation in OVCAR8 on COL6 ($7.2 ± 1.4$-fold increase, $p = 0.0011$), whereas OVCAR4 FA count remained unaltered (Fig. 7e–g). Moreover, in both cells, peripheral translocation of FAs was observed only on COL1, while on COL6 the FA localization in OVCAR8 was comparable to that of untreated cells on COL1 (Fig. 7e–g). On both substrates, cisplatin reduced F-actin fiber anisotropy in OVCAR4, whereas in OVCAR8 it remained constantly low (Fig. 7h). However, cell protrusions, prominent in OVCAR8, were reduced by cisplatin on both COL1 and COL6 (Fig. 7i; $p ≤ 0.014$). Therefore, we further investigated the association between increased central FAs and integrin signaling in OVCAR8 by assessing the downstream phosphorylation of myosin light chain (pMLC), involved in actomyosin contractility and migration[32]. While pMLC was significantly higher in untreated OVCAR8 on COL1 than on COL6, cisplatin increased pMLC on COL6 to similar levels as detected on COL1 (Fig. 7j; on COL6 $1.6 ± 0.0$-fold increase with cisplatin, $p < 0.0001$). Altogether, these results indicate that cisplatin enhances COL6-mediated FA signaling coincident with further increased cisplatin resistance in OVCAR8 cells.

**COL6 confers relapse HGSC patient cells with cisplatin-induced adhesion and cisplatin resistance.** To understand the clinical relevance of the ECM-adhesion-dependent chemoresistance, we isolated HGSC cells from patient ascites pre-chemotherapy (p-HGSC) and at relapse (r-HGSC) followed by ex vivo culture and platinum treatment on COL1 or COL6 (see Supplementary Fig. 15a for the HGSC markers PAX8 and cytokeratin 7 (CK7)[13] and Supplementary Data 1 for patient information). On COL1, the total intensity of active β1-integrin was comparable between the matched patient p-HGSC and r-HGSC cells and remained unaltered by cisplatin (Fig. 8a). On COL6, active β1-integrin was likewise comparable between untreated p-HGSC and r-HGSC, however, cisplatin reduced the activity in p-HGSC, while enhancing the activation twofold in r-HGSC (Fig. 8b; $p ≤ 0.017$). In both these untreated cells, β1-integrin-based FAs were longer on COL6 than on COL1 (Fig. 8c, d; $p ≤ 0.0001$), suggesting increased subcellular adhesion and force transmission[33,34]. In p-HGSC, platinum induced FA elongation on COL1, while shortening FA length on COL6 (Fig. 8c; $p ≤ 0.0002$). Conversely, in r-HGSC cisplatin shortened FAs on COL1 while enhancing FA length on COL6 (Fig. 8d; $p ≤ 0.004$). Despite the larger spreading area of r-HGSC compared to p-HGSC, cisplatin altered the

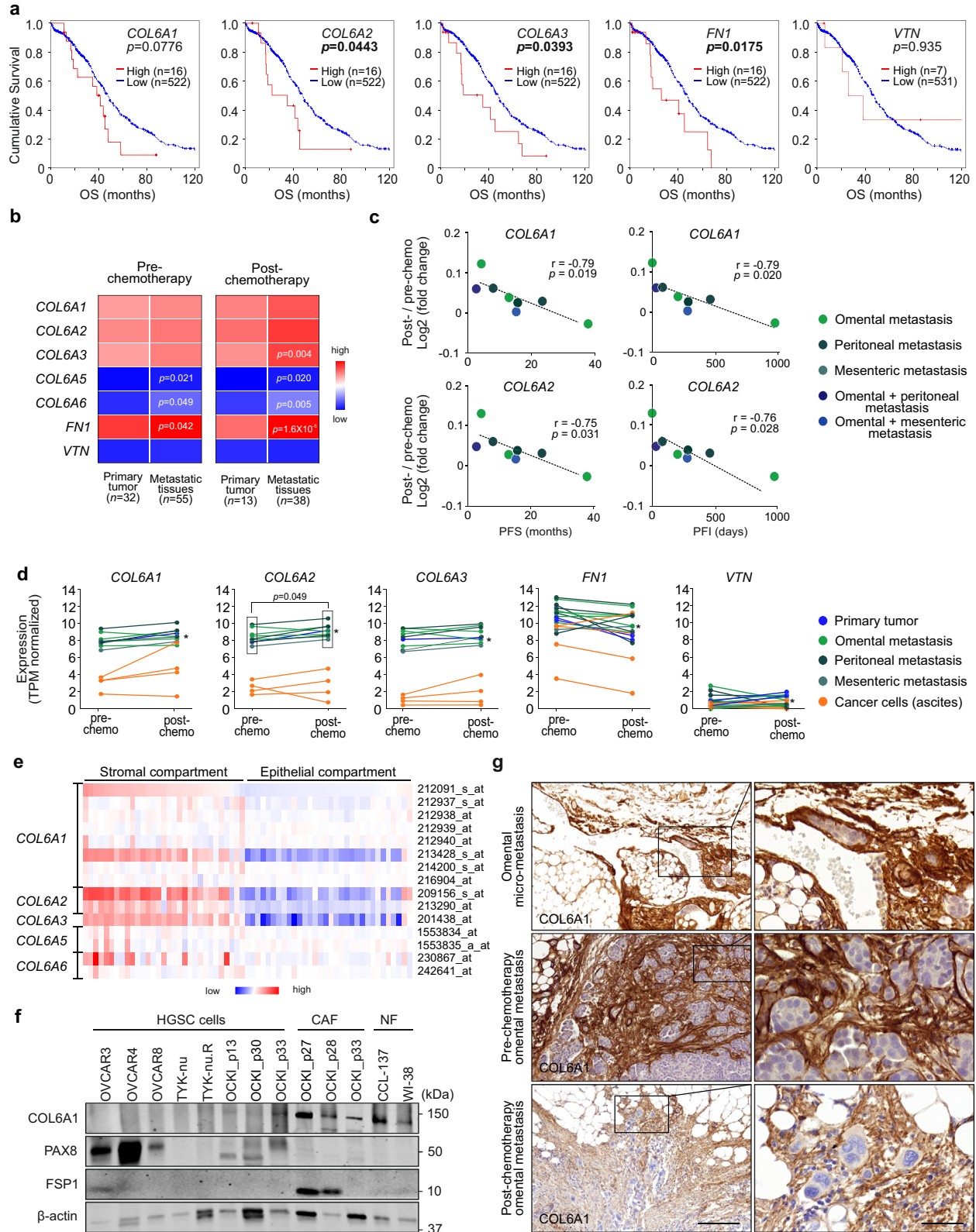

spreading of neither cell type (Supplementary Fig. 15b; p-HGSC $n = 3$, r-HGSC $n = 3$). To investigate whether the diverse β1-integrin adhesion could be related to increased expression of the COL6-binding receptors or cell epithelial–mesenchymal transition (EMT), we analyzed integrin and EMT marker gene expression pre- and post-chemotherapy. Notably, both the integrin and EMT marker expression in ascites-derived cells remained essentially

unaltered (Supplementary Fig. 15c; p-HGSC $n = 9$, r-HGSC $n = 15$), suggesting that the integrin repertoire or EMT state per se do not govern the differential response.

Finally, to assess the HGSC chemoresistance in a tissue-like environment, freshly isolated ascites-derived HGSC cell clusters from patients subjected to either primary debulking surgery (PDS; $n = 4$) or neoadjuvant chemotherapy treatment (NACT;

**Fig. 5 COL6 increases upon chemotherapy and is associated with poor patient survival. a** Kaplan–Meier curves show the association between collagen (COL) *6A1, COL6A2, COL6A3*, fibronectin (*FN1*), and vitronectin (*VTN*) expression in chemo-naïve ovarian cancer tissues with overall survival (OS) in The Cancer Genome Atlas (TCGA) dataset; log-rank test. **b** Heatmap shows average *COL6A1-6A6, FN1*, and *VTN* expression in pre- and post-chemotherapy high-grade serous carcinoma (HGSC) RNA-seq of primary tumor and metastatic (omental+peritoneal+mesenteric) tissue patient samples. Benjamini–Hochberg-corrected *p* value, two-tailed Student's *t* test, *n* = number of samples; the color key indicates the normalized gene expression values (low = 0.00; high = 12.98). **c** Scatter plots depict the correlation of *COL6A1* and *COL6A2* expression fold change (post- against pre-chemotherapy) with platinum-free interval (PFI) and progression-free survival (PFS) in HGSC patient- and tissue-matched samples (*n* = 8); two-tailed Pearson correlation. **d** Charts illustrate change in expression of *COL6A1-A3, FN1*, and *VTN* upon chemotherapy in matched HGSC patient-derived samples (*n* = 12) from initially platinum-sensitive patients (including partial/complete response or stable disease). Asterisk identifies a patient with longer platinum-free interval (974 days) in comparison to other patients (0–460 days). Significance represents the induced expression of *COL6A2* in metastatic tissues post-chemotherapy; two-tailed Student's *t* test. See Supplementary Fig. 12c for corresponding charts for *COL6A5* and *COL6A6*. **e** Heatmap of *COL6A1-A6* gene expression in HGSC cells from stromal (*n* = 31) and epithelial compartments (*n* = 32; GSE40595[29]). Used probe IDs as shown; the color key indicates the normalized gene expression values (low = −5.46; high = 7.10). **f** Immunoblot for COL6A1, PAX8, fibroblast-specific protein 1 (FSP1), and β-actin in HGSC cells, omental metastasis-derived cancer-associated fibroblasts (CAFs), and normal fibroblasts (NFs); *n* = 2 biological replicates. **g** Micrographs of COL6A1 immunohistochemistry reveal abundant protein expression in the immediate surroundings of the malignant HGSC foci in omental micro-metastasis and in pre- and post-chemotherapy omental metastases. Images representative of two (micro-metastasis), six (pre-chemotherapy), and ten (post-chemotherapy) patients. Scale bar = 200 µm and 50 µm in inset. See Supplementary Data 26–29 for specific values (**b–d**). Source data are provided as a Source data file.

*n* = 3) were grown in 3D COL1 or COL1 + COL6 for 4 days, followed by 72-h treatment. After the 4-day 3D culture, these short-term organoids displayed a HGSC morphology, grew invasively in COL1, and were positive for PAX8 and CK7 (Fig. 8e). As with OVCAR8, COL6 reduced relative metabolically active cell content in 2/4 r-HGSC organoids, while in p-HGSC the activity remained unaltered between the matrices or increased in the COL1 + COL6 (Fig. 8f–h; *n* = 8). Most notably, in all the r-HGSC short-term organoids, COL6 conferred cisplatin resistance, whereas p-HGSC were unaffected or even increasingly sensitive to the treatment, regardless of the cell division to prognostic grouping of PDS and NACT treatment arms (Fig. 8f–h; *p* ≤ 0.049). Altogether, these results show that cisplatin enhanced COL6 adhesion, and COL6 increased protection against cisplatin cytotoxicity specifically in r-HGSC cells. Furthermore, these results suggest that the protection by COL6 can derive from intrinsic platinum resistance mechanisms that are already active in the HGSC cells derived from relapse disease.

## Discussion

The recovery and recolonization of treatment-escaping HGSC cells is a major cause of treatment failure[35]. In this study, we present comprehensive matrisome expression signatures of the fibrotic tumors in longitudinal HGSC cohort along with experimental results of matrix component-dependent and stiffness-induced platinum resistance in 2D and 3D cultures and patient-derived organoids. These results provide strong support for the emerging concept that the stiff fibrotic tumor ECM promotes chemoresistance[36,37]. They also demonstrate that chemotherapy alters both the ECM remodeling and sensing, indicating how interrelated the cancer cell intrinsic and TME-dependent chemoresistance mechanisms are. Indeed, the herein observed changes in matrisome, including strong COL6 expression at metastatic sites and upregulation upon treatment, coupled with progressive HGSC changes in the ECM component- and stiffness-dependent β1 integrin-pMLC signaling, explain how tumor evolution can provide unique niches for cancer cells to engage altered ECM remodeling and sensing to drive chemoresistance.

Excessive ECM deposition and remodeling have been observed in HGSC[15,38]. Our systematic description of the previously undefined human HGSC matrisome in primary and metastatic tumors as well as pre- and post-chemotherapy highlights that both chemotherapy-induced changes and the host tissue type determine the tumor ECM gene signatures. General features of the evolving matrisome observed include the expression and

deposition of fibrillar collagens, which leads to tissue stiffening and promotes chemoresistance in various cancer types[39,40]. In line with these reports, our results show that the increase in stiffness, at the range previously detected in HGSC metastases[16], induces FAK- and YAP-dependent resistance to cisplatin-induced accumulation of DNA damage, suggesting implications as a remarkable apoptosis-protecting TME.

Extensively studied in cancer, FN is known to stimulate OC cell proliferation and support cell adhesion and migration via α5β1-integrin/c-MET/FAK/Src pathway[41]. The functionally less studied COL6 has been suggested to act as an anti-apoptotic factor and also to associate with tumor progression and poor survival in solid pre-chemotherapy tumors, including breast and pancreatic cancers[20,42–44]. Consistently, our IHC results reveal COL6- and FN-rich fibrosis already around micro-metastases, which upon disease evolution develops into an extensive desmoplastic TME. Whether the early metastatic cells communicating in omental stroma directly induce this fibrotic response or the pre-metastatic omentum forms fibrotic niches for the metastases, and what are the roles of the other ECM components such as COL5 and COL11 as well as proteolytic enzymes including MMPs also prominent in our matrisome signatures, remain of future interest. Nevertheless, our results indicate that stromal COL6 is well positioned to support growth and apoptosis evasion in early metastasis and most importantly that COL6-rich ECM encapsulates the treatment-escaping, residual micro-metastatic HGSC lesions.

Previously, upregulation of *COL6A3* has been reported in platinum-resistant OC cells in vitro, and inhibition of the cleaved product of COL6A3, endotrophin, can sensitize cisplatin-resistant breast cancer cells[45,46]. Yet, the cellular source and mechanisms how COL6 contributes to chemoresistance in HGSC remain unclear. Our current results strongly suggest that the tumor stroma is the major source of COL6, which increased gene expression in the solid tumors correlates with decreased PFI and PFS. Further, in patients who developed chemotherapy resistance, *COL6A2* increased upon chemotherapy treatment in solid tumors, revealing its potential as a marker predicting poor chemotherapy response and shortened survival. Although at clearly lower levels, *COL6* expression increased also in individual ascites cells. It also remains possible that adherent HGSC cells in solid tumors express more ECM than those in suspension, raising the possibility that besides the stroma some cancer cells may upregulate *COL6*, particularly upon the development of resistant disease.

Mechanistically, current results reveal an unexpected platinum-induced, COL6-dependent migratory response coincident with

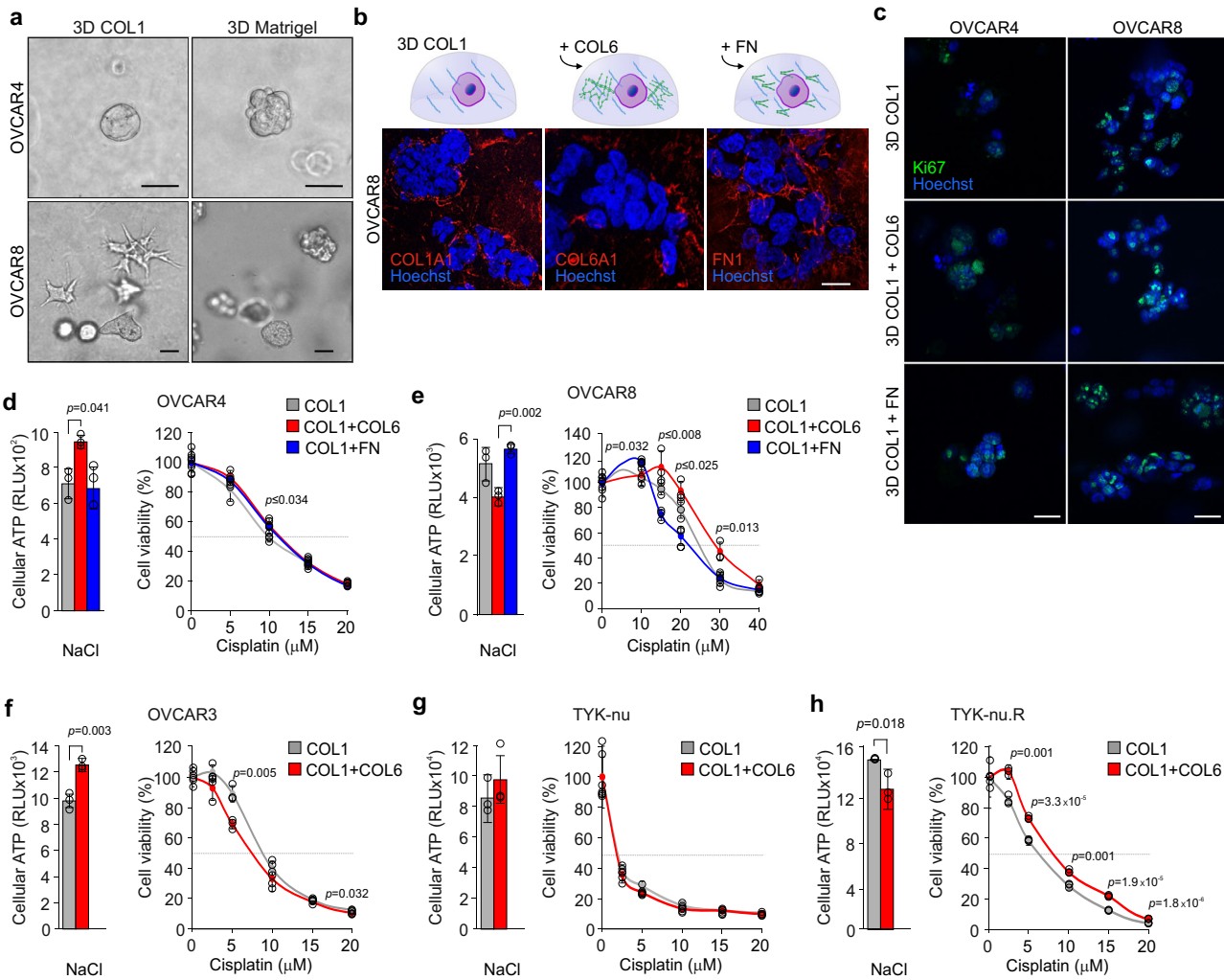

**Fig. 6 Addition of collagen 6 promotes HGSC cell survival in 3D collagen 1. a** Representative light phase-contrast images show the diverse growth phenotype response of OVCAR4 and OVCAR8 cultured in 3D collagen 1 (COL1) and Matrigel for 5 days. Scale bar = 25 μm. **b** Schematic diagram of the experimental design and representative confocal micrographs of COL1A1, COL6A1, and fibronectin (FN1) in OVCAR8 grown for 5 days in 3D COL1 with or without 50 μg/ml COL6 or FN supplementation. Scale bar = 25 μm. **c** Representative confocal micrographs of Ki67 (green) show the comparable proliferation in OVCAR4 and OVCAR8 between all corresponding 3D matrix cultures. Cells were grown for 5 days. See Supplementary Fig. 13b, c for quantifications. Scale bar = 25 μm. **d–h** Bar charts depict cellular ATP in OVCAR4 (**d** COL1 vs COL1 + COL6 $p = 0.016$ vs COL1 + FN $p = 0.034$), OVCAR8 (**e** COL1 + COL6 vs COL1 + FN at 10 μM $p = 0.005$, at 20 μM $p = 0.004$, at 30 μM and vs COL1 $p = 0.013$; COL1 vs COL1 + FN at 10 μM $p = 0.008$, at 20 μM $p = 0.025$), OVCAR3 (**f**), TYK-nu (**g**), and TYK-nu.R (**h**) grown in corresponding 3D matrices for a total of 5 days. Scatter plots illustrate cell viability after 72 h treatment with 0–20 μM (**d**, **f–h**) or 0–40 μM (**e** OVCAR8) cisplatin. Cell viability was determined by ATP measurement and is shown in reference to NaCl (control) per matrix. Data represent mean ± SEM of biological replicates ($n = 3$, OVCAR4 in **c**, **d–h** $n = 4$ in OVCAR8 in **c**); two-tailed Student's *t* test. Images in **a**, **b** are representative of 6 biological replicates. Source data are provided as a Source data file.

enhanced COL6-mediated platinum resistance and a change in cell adhesion signaling via the stiffness-dependent β1 integrin-pMLC and YAP/TAZ pathways. Changes in adhesion can derive from alterations in actin cytoskeleton structure and contractility[47–49]. Despite these reports pointing toward decreased actomyosin activity, we show that, in cells with COL6-induced chemoresistance, cisplatin enhances myosin activation. Although the reciprocal regulation of FA formation and acto-myosin contractility makes the identification of the primary cisplatin molecular responder challenging, this enhanced activation likely relates to the observed increase in β1 integrin activation and FA formation.

The diverse cancer cell responses described in this study to ECM proteins could support the need of precision medicine approaches to target the ECM pathways and their effects on disease development and chemoresistance. Nevertheless, the strong impact of

stiffness on resistance, coupled to specific effect of COL6 on relapse HGSC patient-derived cells, suggests the possibility of a broader application of biomechanics signaling or COL6-based targeted therapies in relapse patients with highly fibrotic tumors. While integrin blocking combined with chemotherapy has not shown successful results in patients[50], other approaches targeting adhesion signaling as well as COL6 are under investigation. Such approaches include small molecule inhibitors of ATK and FAK and antibody-based targeting against cleaved C5A fragment of COL6[46,51]. However, further mechanistic understanding of the differential responses of chemotherapy-naïve and relapsed disease cells will provide a better interpretation whether the potential of COL6 as a therapeutic target relies on targeting the ECM stiffness-induced dysregulated adipocytes depositing COL6[52], upstream pathways in CAFs secreting COL6[53,54], or in the infiltration of other cell types modifying the ECM.

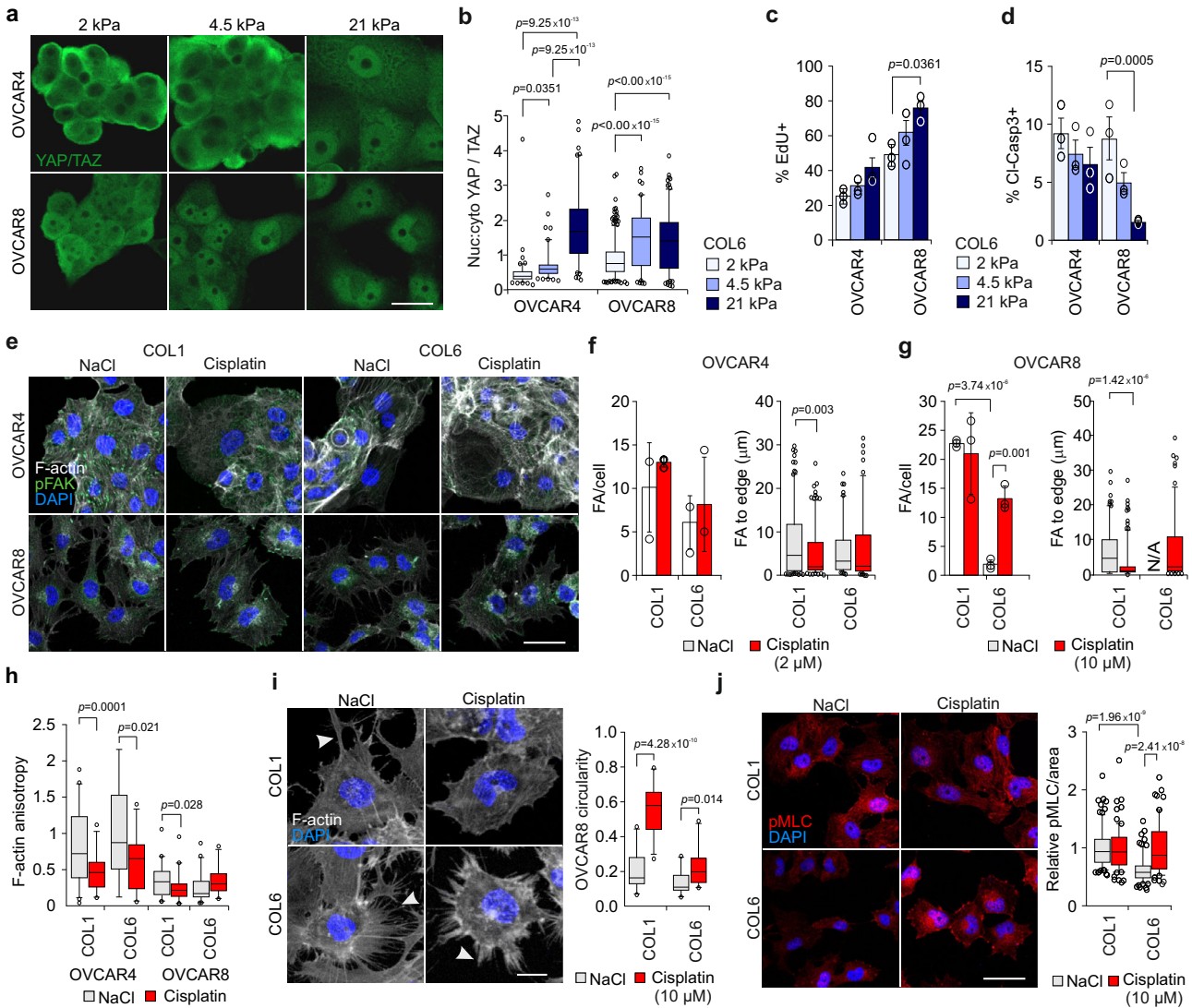

**Fig. 7 Cisplatin treatment enhances integrin-based cancer cell adhesion on stiff collagen 6 substrate. a, b** Representative confocal micrographs (**a**) and corresponding nuclear to cytoplasmic ratio (nuc:cyto; **b**) of YAP/TAZ (green) in cells on 2, 4.5, and 21 kPa collagen 6-functionalized polyacrylamide hydrogels (COL6-PAA) at 48 h; OVCAR4 $n = 114/111/103$ and OVCAR8 374/265/165 cells, respectively; Scale bar = 10 μm. **c, d** Charts depict positivity for EdU in untreated (**c**) and cleaved caspase3 in treated (cl-Casp3; **d**) cells on corresponding COL6-PAA at 24 h; see Supplementary Fig. 14c, d for representative micrographs. **e–g** Representative confocal micrographs of F-actin (phalloidin, white) and phosphorylated focal adhesion (pFAK, green) in cells on 21 kPa COL1 or COL6-PAA at 48 h. Charts show the number and peripheral localization of FAs in OVCAR4 (**f** $n = 186/191$ and 103/105) and in OVCAR8 (**g** $n = 118/195$ and not applicable (N/A)/103). Scale bar = 25 μm. **h** Chart depicts the F-actin anisotropy in cells on 21 kPa COL1 or COL6-PAA at 48 h; OVCAR4 $n = 47/35$, 18/30 and OVCAR8 $n = 44/41$, 46/27 cells, respectively. See Fig. 7e for representative micrographs. **i** Representative confocal micrographs of F-actin (phalloidin, white) in OVCAR8 on 21 kPa COL1 and COL6-PAA at 24 h; $n = 25$ cells. White arrows indicate protrusions. Scale bar = 10 μm. **j** Representative confocal micrographs of myosin light chain (pMLC, red) in OVCAR8 on 21 kPa COL1 and COL6-PAA at 24 h. Chart indicates pMLC staining intensity per area; $n = 86/72$ and 75/65 cells, respectively. Scale bar = 25 μm. Data represent mean ± SEM of biological replicates ($n = 3$ in **b, d–j**, $n = 4$ in **c**). One-way ANOVA with Tukey's multiple comparison test (**b, f, g, i, j, l**); two-tailed Student's $t$ test (**c, d, h**). Box plots indicate median (middle line), 25th, 75th percentile (box), and 10th and 90th percentile (whiskers) as well as outliers (single points). Source data are provided as a Source data file.

In conclusion, we provide extensive evidence that ECM biochemical properties and biomechanical signaling are critical factors in cancer cell survival and chemotherapy resistance. As the stromal compartment is essential in mediating tumor progression and chemotherapy resistance, future mechanistic investigations targeting the COL6-mediated and/or stiffness-dependent signaling should be explored.

## Methods

**Antibodies and reagents**. The following antibodies and reagents were used: CD45 (clone 2B11 + PD7/26, Dako, IHC 1:100), CD68 (Sigma-Aldrich, IHC 1:1000),

CK7 (ImmunoWay, immunofluorescence (IF) 1:600 (2D), IF 1:200 (3D), IHC 1:100); COL1A1 (Abcam, IF 1:100, IHC 1:1000); COL6A1 (Abcam, IF 1:100, IHC 1:1000; clone B-4, Santa Cruz, immunoblot (IB) 1:1000); FN1 (Sigma-Aldrich, IF 1:300, IHC 1:1000); fibroblast-specific protein 1 (FSP1; S100A, Proteintech, IB 1:750); phosphorylated (S139) gamma H2AX (Cell Signaling Technologies, IF 1:200; Abcam IF 1:400); RAD51 (Abcam, IF 1:400), CyclinA2 (Gene-Tex, IF 1:400), active-integrin β1 (clone 12G10, Abcam, IF 1:400); PAX8 (Proteintech, IF and IHC 1:100, IB 1:2500); Phalloidin-TRICT conjugated (Sigma-Aldrich, IF 1:40); Phalloidin AlexaFluor 647-conjugated (ThermoFisher, IF 1:100); Ki67 (NCL-Ki67p, Leica Biosystems, IF 1:1500, IHC 1:3000); cleaved caspase-3 AlexaFluor488-conjugated (Cell Signaling Technologies, IF 1:700); cleaved caspase 3/7 (CellEvent Green Detection Reagent, Invitrogen, IF 2 μM); phosphorylated (Y397) FAK (BD Biosciences, IF 1:200); phosphorylated (S20) myosin light chain (pMLC; Abcam, IF

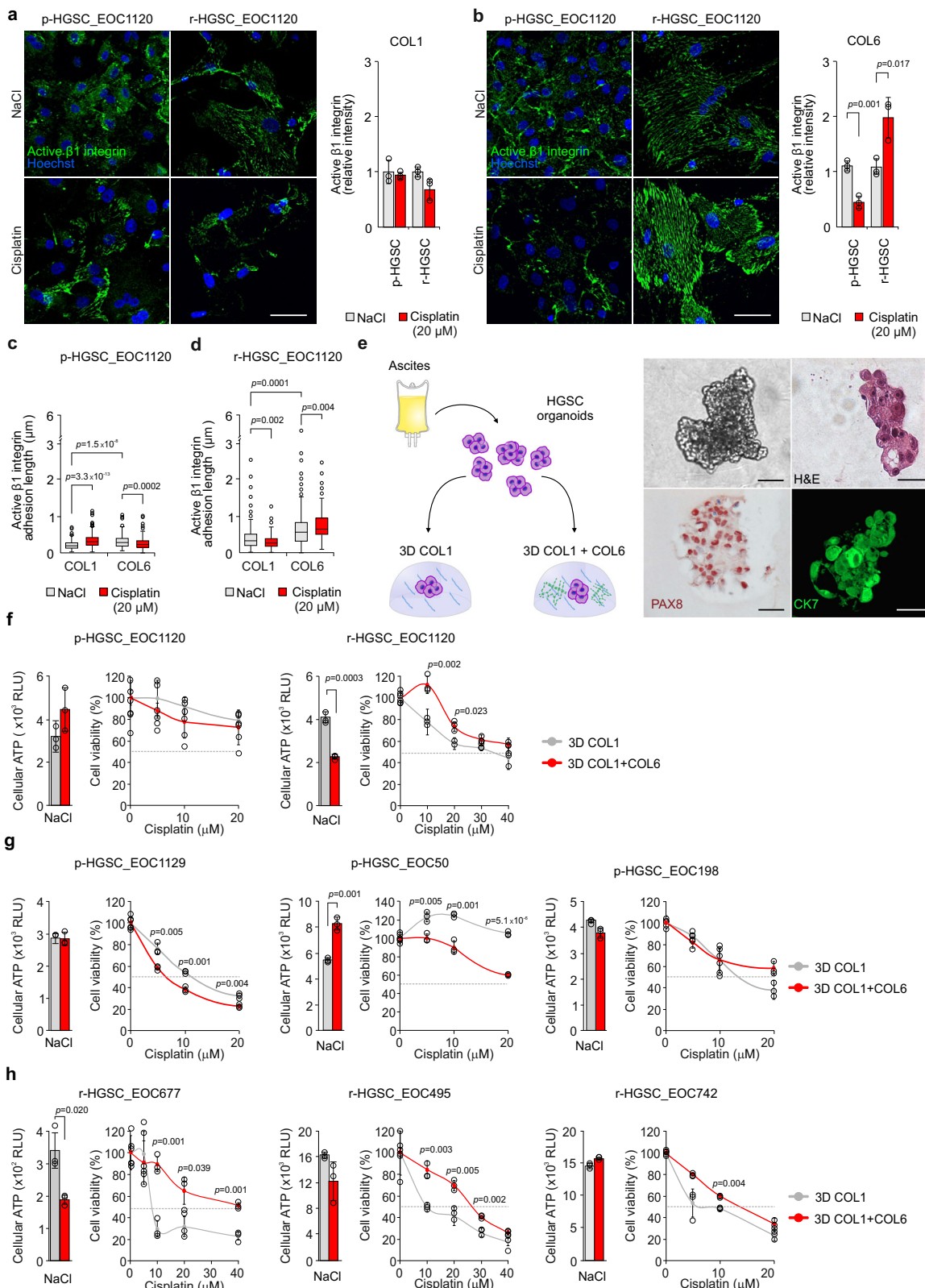

1:400); YAP/TAZ (63.7, Santa Cruz, IF 1:200); Hoechst #33342 (Thermo Scientific, IF 10 μg/ml); β-actin (clone C-4, Santa Cruz, IB 1:2000). For IF, AlexaFluor-conjugated (Invitrogen) secondary antibodies were used. For mounting, Vecta-shield with 4',6-diamidine-2-phenylindole dihydrochloride (DAPI; H-1200, Vector Laboratories) or without (H-1000) were used. For IHC, hematoxylin solution according to Mayer (Sigma-Aldrich), Eosin Y (Merck/Millipore), and horseradish peroxidase-conjugated secondary antibodies (DAKO, Agilent Technologies) were used. Other reagents used were: Click-iT™ Plus Edu Alexa Fluor 647 Imaging Kit (Invitrogen), acid-extracted rat tail collagen type 1 (Sigma-Aldrich), human recombinant collagen type 1 (Rockland), human collagen type III (Advanced Biomatrix), human collagen type IV (CellSystems), human collagen type VI (Rockland), human plasma-derived FN (Sigma-Aldrich), human VTN (CellSystems), human recombinant tropoelastin (Advanced Biomatrix), growth factor reduced Matrigel (Corning), natrium chloride (NaCl, Fresenius, Kabi, AG), Defactinib (Selleckchem), Verteporfin (Sigma-Aldrich), and cisplatin (Sigma-Aldrich).

**Fig. 8 Collagen 6 confers relapse HGSC patient cells with cisplatin-induced adhesion and cisplatin resistance. a–d** Representative light micrographs of active β1 integrin (green) in EOC1120 patient ascites-derived cells collected pre-chemotherapy (p-HGSC) at primary debulking surgery (PDS) and at relapse (r-HGSC) stages and grown for 24 h on 3D collagen 1 (COL1) and COL6 before receiving NaCl (control) or 20 μM cisplatin for 48 h. Charts illustrate active β1 integrin intensity (**a**, **b**) and adhesion length (**c**, **d**); p-HGSC 205/255 and 180/289 cells, respectively; r-HGSC 349/360 and 343/311 cells, respectively, within $n = 3$ biological replicates. Scale bar (**a**, **b**) = 50 μm. **e** Schematic diagram of the experimental design and representative micrographs of p-HGSC organoids obtained from patient EOC1032 at interval debulking surgery after neoadjuvant chemotherapy treatment (NACT) and grown in 3D matrices for 5 days. Phase-contrast light micrograph and hematoxylin–eosin (H&E) staining depict organoid phenotype and morphology. Immunohistochemistry and immunofluorescence show PAX8 and cytokeratin 7 (CK7, green) positivity; $n = 4$ biological replicates. Scale bar = 25 μm. **f–h** Bar charts depict cellular ATP in EOC1120 (PDS) p-HGSC (pre-chemo)/r-HGSC (post-chemo) (**f**) and 3 unpaired p-HGSC (**g** pre-chemo, EOC1129, EOC50 from NACT, EOC198 from PDS) and r-HGSC (**h** post-chemo, EOC677 from NACT, EOC495 and EOC742 from PDS) organoids grown for a total of 7 days in 3D COL1 or COL1 + COL6 (50 μg/ml). Scatter plots illustrate cell viability after 72 h treatment with 0-40 μM cisplatin, which was determined by ATP measurement and is shown in reference to NaCl (control) per matrix; $n = 3$ biological replicates. Data represent mean ± SEM; two-tailed Student's $t$ test (**a**, **b**, **f–h**); one-way ANOVA with Tukey's multiple comparison test (**c**, **d**). Box plots indicate median (middle line), 25th, 75th percentile (box), and 10th and 90th percentile (whiskers) as well as outliers (single points). Source data are provided as a Source data file.

**Patient material and clinical data**. All studies involving clinical material were performed in accordance with the ethical standards from the 1975 Declaration of Helsinki. Each patient gave written informed consent and the use of HGSC clinical material was approved by the Ethics Committee of the Hospital District of Southwest Finland (ETMK), The National Supervisory Authority for Welfare and Health (Valvira), and The Swedish Ethical Review Agency (Etik-prövningsmyndigheten). Control omentum tissue sections were collected under the auspices of Auria Biobank.

HGSC samples collected at the Turku University Hospital were from patients undergoing PDS ($n = 18$) or NACT followed by interval debulking surgery (IDS) if estimated inoperable in diagnostic laparoscopy ($n = 45$; see Supplementary Data 1 and 2 for detailed information). Patient tissues (primary ovary/fallopian tube tumors; omental, peritoneal, and mesenteric metastases) and ascites were collected from pre-chemotherapy (diagnostic laparoscopy or PDS) and post-chemotherapy (IDS) surgeries and from palliative puncture at relapse (ascites only).

For RNA sequencing (RNA-seq), total RNA was extracted from fresh patient tissues and ascites-derived cells by using the RNeasy Kit from QIAGEN (Qiagen, Hilden, Germany) with DNase I treatment used according to the manufacturer's protocol. RNA quality and concentration were tested with Bioanalyzer 2100 (Agilent, CA, USA). All samples fulfilled the criteria of RNA integrity number >7 and concentration >50 ng/μl.

For the ex vivo experiments, the ascites fluid was received in a sterile secretion bag and processed within 30 h from the operation or puncture and both from patients undergoing NACT (EOC1032, EOC1129, EOC198, EOC677, EOC26, EOC691, and EOC167) or PDS (EOC742, EOC1120, EOC50, EOC495) treatment arm. The ascites fluid was centrifuged at $3200 \times g$ for 10 min at 4 °C for obtaining clarified ascites, which was stored at −70 °C. From the cell pellets, erythrocytes were lysed using Tris-buffered ammonium chloride solution (Tris-NH$_4$Cl) and multicellular clusters were collected by using 40-μm-pore-sized strainer (pluriSelect). Cell clusters were cultured in Dulbecco's modified Eagle's medium (DMEM)/F-12 (Gibco) supplemented with 20% patients' clarified ascites, 1% penicillin and streptomycin (pen–strep), and 10 mM HEPES (Gibco). Within 24 h from starting the culture, multicellular clusters were characterized as CK7 and PAX8 positive by performing IF from Cytospin samples. Embedding of cell clusters in 3D was done within 7 days after receiving the samples.

Abdominal ascites fluid and omental tumors were also collected at the Karolinska University Hospital (Sweden) as previously described[19]. Cancer cells obtained from ascites fluid and CAFs derived from omental metastases used in this study are indicated as OCKI. For pre-chemo OCKI patient information, refer to Moyano-Galceran et al.[19] and Supplementary Data 1 for OCKI_p30 and OCKI_p33.

**RNA-seq and bioinformatics**. Sequencing libraries were constructed in Beijing Genomics Institute (Beijing, China) using a modified protocol similar to the TrueSeq Stranded Total RNA with RiboZero Kit (Illumina Inc.). Paired-end 100 bp RNA-seq producing around 60 M reads was carried out on Illumina HiSeq4000 and HiSeq X-Ten and BGISEQ-500 platforms. The data were processed using SePIA, a comprehensive RNA-seq data processing workflow[55]. Read pairs were trimmed using Trimmomatic[56] (version 0.33) as follows: (i) the first 12 and last 5 bases were cropped due to uneven per base sequence content; (ii) any leading bases with a quality score <20 and any trailing bases with a quality score <30 were removed; (iii) the reads were scanned with a 3-base wide sliding window, cutting when the average quality per base dropped <20; (iv) resulting sequences <20 bp were discarded. Trimmed reads were aligned to the reference genome (GRCh38.d1. vd1) using STAR[57] (version 2.5.2b), allowing up to 10 mismatches, and all align-ments for a read were output. Gene-level expression was quantified as $\log_2(\text{TMP} + 1)$, where TPM is transcript per million as calculated by eXpress[58] (version 1.5.1-linux_x86_64). For the TPM analysis that consisted of multiple detections per sample (i.e. seqXa, and seqXb), an average value was used. For the analysis of DEGs, DESeq2[59] (version 1.22.1) with default settings was used on raw counts to call DEGs based on 11 comparisons, including different anatomical locations and chemotherapy phases. For the DEG analysis of resistant and sensitive platinum disease, patients were grouped by their PFI as resistant (PFI ≤ 6 months (180 days)) or as sensitive (PFI > 6 months (181 days)). Ascites-derived cancer cells from progressive disease were excluded from this analysis due to their biasness for platinum resistance. $p$ Values were adjusted for multiple hypothesis testing for 1027 specific mRNAs of our interest using Benjamin-Hochberg method[60]. The significant genes were considered to have adjusted $p$ value of <0.05 and fold change ($\log_2$) $\le -1.00$ or ≥1.00. Relative proportion of DEGs is shown in relation to the presentation of corresponding category in the group of 1027 genes encoding matrisome proteins. Genes CCL4L1 and MUC8 were excluded from the analysis due to missing distinctive Ensembl code.

For enrichment analysis, DEG data were analyzed using IPA[61] (QIAGEN Inc., https://www.qiagenbioinformatics.com/products/ingenuity-pathway-analysis). Direct or indirect relationships were identified with path explorer tool through the ingenuity knowledge base with the most stringent confidence level (experimentally observed; $-\log_{10}(p) > 1.3$, Fisher's exact test).

**Cell lines**. Human HGSC cell lines[21] OVCAR3, OVCAR4, and OVCAR8 (National Cancer Institute, USA) were maintained in RPMI; TYK-nu and TYK-nu.R cells (Japanese Collection of Research Bioresources Cell Bank, Osaka, Japan) were main-tained in Minimum Essential Medium (MEM); and NFs CCL-137 and WI-38 (American Type Culture Collection, USA) were maintained in DMEM; media were supplemented with 10% fetal bovine serum (FBS; Gibco), 1% pen–strep (Gibco), and 1% glutaMAX (Gibco). Cell lines were not authenticated. Cells were cultured according to the manufacturer's instructions and checked routinely for mycoplasma contamination using the MycoAlertPlus Mycoplasma Detection Kit (Lonza).

**Polyacrylamide gel preparation**. To prepare polyacrylamide gels, glass coverslips were washed twice with 70% ethanol, activated with 0.1 M NaOH, and silanized with (3-aminopropyl) trimethoxysilane (Sigma). Then the glass coverslips were treated with 0.5% glutaraldehyde (Sigma) for 30 min and washed extensively with MilliQ water. Mixtures of MilliQ water, acrylamide monomers (Sigma), and crosslinker N,N-methylene-bis-acrylamide (Sigma) were prepared according to previously determined formulations[62,63]. For the polymerization reaction, 5 μl of 10% ammonium persulfate (Sigma) and 0.75 μl N,N,N′,N′-tetramethylethylenediamine (Sigma) were added into 0.5 ml mixtures. To functionalize gels with collagen, gels were first treated with 1 mg/ml N-sulfosuccinimidyl-6-(4′-azido-2′-nitrophenylamino) hexanoate (Sigma), which was activated by ultraviolet (UV) light. Finally, gels were incubated with 10 μg/ml collagen I from rat tail (Sigma) or human collagen VI (Rockland) for 3 h at room temperature (RT) and UV-sterilized prior to cell seeding.

**ECM array**. OVCAR4 and OVCAR8 cell lines were plated at cell density of $5 \times 10^4$ cells/ml onto the ECM Select Array Kit (Advanced Biomatrix) containing 36 different single ECM protein or combinations, each in 9 micro-spots, for 24-h adherence followed by 48-h incubation with or without cisplatin at the concentration of 2 μM for OVCAR4 and 10 μM for OVCAR8. After the 72-h incubation, cell number was observed. Following the manufacturer's instructions, cells were fixed with cold 4% paraformaldehyde first 5 min at +4 °C, then at RT for 10 min. Slides were washed with 1× Hanks' Balanced Salt Solution (HBSS, Sigma), and cells were stained immunofluorescently with phalloidin and Hoechst as described below. For TYK-nu and TYK-nu.R cell lines, wells of opaque-walled 96-well plates (Greiner) were coated with total of 250 μg/ml COL1, COL3, COL4, COL6, FN, ELN, LAM, and VTN as a single ECM protein or combination substrate overnight at +4 °C and 60 min at +37 °C followed by 3× washes with 1× phosphate-buffered saline (PBS). TYK-nu and TYK-nu.R cells were plated at a cell density of $5 \times 10^4$ cells/ml and allowed to adhere for 24-h before treating with cisplatin at a concentration of 2 μM for TYK-nu and TYK-nu.R for 48 h. After the 72-h incubation, cell viability was determined by CellTiter-Glo® Luminescent (Promega).

For 2D 1% serum ECM array, 96-well opaque-well plates were coated with 50 µg/ml COL1, COL6, FN and VTN overnight at +4 °C and 60 min at +37 °C, followed by bovine serum albumin (BSA; 0.1%) blocking for 60 min at +37 °C followed by 3× washes with 1× PBS. Cells were plated at a density of $5 \times 10^4$ cells/ml with full growth media supplemented with either 1% or 10% FBS and allowed to adhere for 24 h before treating with cisplatin at the concentration of 2 µM for OVCAR3, OVCAR4, TYK-nu, and TYK-nu.R and 10 µM for OVCAR8 for 48 h. Cells were phase contrast imaged by IncuCyte S3 Live-Cell Analysis System using the IncuCyte 2020B software (Sartorius) at 24, 36, and 72 h to determine the cell count. After the incubation, cell viability was determined by CellTiter-Glo® as described below.

Cell response was determined as follows: [cell count or cell viability (NaCl) − cell count or cell viability (cisplatin)]/cell count or cell viability (NaCl).

**Live cell imaging**. Polyacrylamide gels were prepared in a silanized glass-bottom 24-well plate (MatTek) by adding 5 µl acrylamide/bis-acrylamide solution onto the glass and covering it with a Rain X-treated 8 mm diameter glass coverslip until polymerization was completed. The gels were activated and functionalized with collagen I as above. HGSC cells were seeded as 12,000 cells/well and allowed to adhere overnight in complete medium. Subsequently, medium was exchanged by phenol red-free media containing 15 mM HEPES (Sigma), cisplatin, and 2 µM CellEvent™ Caspase-3/7 Green (Invitrogen). Cells were imaged (brightfield and green channels) every 3 h for a total of 72 h.

**Cytotoxicity**. For cisplatin cytotoxicity, the end-point cell viability was determined by measuring total cellular ATP by CellTiter-Glo® Luminescent Cell Viability Assay (Promega). For assessing the cell viability, manufacturer's protocol was followed. Briefly, cell cultures with cisplatin treatment were equilibrated for 30 min at RT and CellTiter-Glo reagent was added in 1:1 volume. After inducing the cell lysis by mixing the well contents on an orbital shaker for 5 min (2D) or 10 min (3D), luminescence was stabilized by a 10-min incubation at RT. Luminescence was recorded by FLUOstar® Omega (BMG LABTECH).

**3D cultures**. Collagen-rich 3D hydrogel cultures were prepared by dissolving rat tail collagen type I (Sigma-Aldrich) in 0.25% acetic acid to 4.5 mg/ml concentration at +4 °C and diluting it 1:1 with 2× MEM (Thermo-Fisher) followed by neutralization with NaOH[64]. Hydrogel cell suspensions were left to solidify at +37 °C for 40 min before adding cell culture medium. When indicated, collagen I hydrogel solution was supplemented with 50 µg/ml recombinant human collagen VI (Rockland) or human recombinant FN (Sigma-Aldrich). Laminin-rich hydrogels were prepared by diluting growth factor reduced Matrigel (20.67 mg/ml, Corning) with HBSS (Sigma-Aldrich) to 12 mg/ml concentration. Cultures were imaged with an inverted epifluorescence microscope with a Plan-Neofluar ×5, numerical aperture = 0.15 objective (Axiovert 200; Carl Zeiss, Jena, Germany).

**2D and 3D whole-mount IF**. Cells were fixed with 4% PFA for 20 min (2D) or 60 min (3D) and briefly post-fixed in methanol–acetone (1:1) or absolute ethanol (for monoclonal antibody PAX8 staining) and subjected to blocking and permeabilization with 5% BSA (2D) or 15% FBS (3D) in PBS containing 0.1% Triton X-100 for 30 min (2D) or 0.3% Triton X-100 for 2 h (3D) at RT. Pre-titrated dilutions of primary antibodies were incubated overnight at +4 °C in blocking buffer, followed by thorough washes with PBS (2D) or PBS containing 0.45% Triton X-100 (3D) and incubation with Alexa-conjugated anti-mouse immunoglobulin (Ig), anti-rabbit Ig, anti-goat Ig, anti-Rat Ig, or streptavidin secondary antibodies for 30 min (2D) or 4 h (3D). Cells grown on 2D were subsequently washed and mounted with Vectashield containing DAPI, whereas cells grown in 3D and in ECM array were stained with 5 µg/ml Hoechst diluted in Dulbecco-PBS (Gibco), incubated for 30 min, washed with PBS, and mounted with Vectashield without DAPI.

**Immunohistochemistry**. Sections of healthy omental tissues and malignant HGSC tissues were stained against COL1A1 (Abcam, 1:1000), COL6A1 (Abcam, 1:1000), and FN1 (Sigma-Aldrich, 1:1000) using automated immunostaining device BenchMark XT (Roche Diagnostics/Ventana Medical Systems, Tucson, AZ, USA) using the ultraview Universal DAB Detection Kit (Roche Diagnostics/Ventana Medical Systems) following the manufacturer's manual. Sections of malignant HGSC tissues were stained for CD68 (Sigma-Aldrich, 1:1000) and CD45 (Dako, 1:100), and sections of 3D COL1-embedded HGSC patient cancer cells were stained for CK7 (ImmunoWay, 1:100), PAX8 (Proteintech, 1:100), and hematoxylin–eosin as follows: sections were deparaffinized using TissueClear (Tissue-Tek, Sakura Finetek) and rehydrated in a graded ethanol series. After 20 min pH 6.0 citrate antigen retrieval, sections were incubated with 0.6% (v/v) hydrogen peroxide for 10 min. Pre-titrated dilutions of primary antibodies were incubated overnight at +4 °C in 0.1 M Tris-HCl pH 7.5, 0.15 M NaCl, and 0.5% TSA blocking reagent (FP1020, Perkin Elmer, MA, USA), followed by incubation with biotinylated secondary antibodies, TSA amplification (Perkin Elmer), and chromogenic detection with 3-amino-9-ethylcarbazole. Stainings were imaged using Leica DM LB microscope or 3DHISTECH Pannoramic 250 FLASH II digital slide scanner equipped with the software Pannoramic Viewer 2.0 - 250.

**Immunoblotting**. HGSC cell lines, OCKI patient-derived cancer cells and CAFs, as well as NFs were characterized by IB for COL6A1, PAX8, and FSP1. Cells were lysed with RIPA (50 mM Tris-HCL pH 7.4, 150 mM NaCl, 1% Igepal CA630, 0.5% sodium deoxycholate) buffer together with 5 mM EDTA, protease inhibitor (cOmplete ULTRA, Sigma), and phosphatase inhibitor (phosSTOP, Sigma) and incubated for 20 min on ice. Cell lysates were centrifuged at 21,300 × g for 15 min at +4 °C. After mixing the lysates with 5× sample buffer (0.3 M Tris-HCl pH 6.8, 50% glycerol, 10% sodium dodecyl sulfate, 0.05% bromophenol blue; 0.5 M dithiothreitol), heat denaturation was done by incubation at +95 °C for 10 min. Lysates were then separated in 4–20% Mini-PROTEAN TGX Precast Protein Gels (Bio-Rad) and transferred to Trans-Blot Turbo Mini Nitrocellulose Transfer Packs (Bio-Rad). Membranes were blocked for 45 min with 3% fish gelatin (Sigma) in Tris-buffered saline (TBS; 10 mM Tris, pH 7.6, 150 mM NaCl) and probed with primary antibody in TBS 0.1% Tween-20 (TBS-T) with 3% fish gelatin at the recommended dilutions at +4 °C overnight. Membranes were incubated with IRDye Subclass-Specific Antibodies (LI-COR Biosciences) diluted in TBS-T for 1 h at RT, and the signal was detected using Odyssey Imaging System equipped with the software Image Studio Lite Ver 5.2 (LI-COR Biosciences).

**Image acquisition and quantifications**. Live cell imaging was done by using Cytation 5 imaging reader at +37 °C and 5% $CO_2$ (BioTek™ CYT5MPV) using the ×10 objective. Light micrographs were taken using Confocal Zeiss LSM 780 and Zeiss AxioImager.Z1 upright epifluorescence microscopes with Apotome combined with a computer-controlled Hamamatsu Orca R2 1.3 megapixel monochrome CCD camera and ZEN software (ZEN 2.3 and 2.6 Blue as well as ZEN 2.3 Black edition (Zeiss)). ×40 Plan Apochromat, 1.4 NA Oil objective was used. Post-acquisition image processing was performed using the Corel Draw software X5. Brightness and contrast were linearly adjusted using Corel Photo-Paint X8. IF and live cell imaging-based quantifications were done by using CellProfiler (3.1.8)[65], Qupath software (v0.1.2)[66], and ImageJ (1.52p) with FibrilTool plug-in[67].

**Statistical analysis**. All numerical values represent mean ± SEM unless stated otherwise. The data distribution was tested by Kolmogorov–Smirnov and Shapiro–Wilk normality tests together with histogram analyses. Statistical significances were determined using two-tailed Student's t tests, Mann–Whitney U, one-way analysis of variance with Tukey's multiple comparison test or log-rank test as indicated in the figure legends. For correlation analyses, Pearson correlation coefficient was used. For transcriptome analysis, the Benjamin–Hochberg multiple comparison adjustment was applied and the corrected p value <0.05 was considered as statistically significant. For the analysis of publicly available serous ovarian carcinoma patient cohort, the TCGA dataset was obtained from http://cancergenome.nih.gov/ and HGSC stroma/epithelial tumor dataset from gene expression omnibus (GSE40595)[29]. Analysis of TCGA was done based on the mRNA data (Agilent microarray; n = 538) dataset. Survival functions were estimated with Kaplan–Meier method and compared using log-rank test. Statistical analyses were performed using IBM SPSS Statistics 25 and Graphpad Prism 7.

**Reporting summary**. Further information on research design is available in the Nature Research Reporting Summary linked to this article.

## Data availability

All processed RNA sequencing data used for the analysis in this study (Figs. 1b–e, 2a–e, 5b–d and Supplementary Figs. 1a, b, 2, 3b–e, 12b, c, 15c) are available in public Gene Expression Omnibus (GEO) database under the accession code GSE173420. The corresponding raw RNA sequencing data are available in the European Genome-Phenome Archive (EGA) with the accession code EGAD00001006456, under the study EGAS00001004714. Restrictions due to EU General Data Protection Regulation (GDPR) forbid granting access to anonymous persons. Thus, the raw data are available upon request from the Data Access Committee at sysbio-dac@helsinki.fi. The dataset of 32 epithelial HGSC and 31 stromal samples are publicly available at GEO under accession code GSE40595 (Fig. 5e). The Cancer Genome Atlas ovarian serous cystadenocarcinoma patient dataset (TCGA, Firehose Legacy) is available at https://portal.gdc.cancer.gov/projects/TCGA-OV (Fig. 5a and Supplementary Fig. 12a). The remaining data are available within the Article and Supplementary Information. Source data are provided with this paper.

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

## Acknowledgements

We thank Stephen J Weiss and Jaana Oikkonen for comments, Anastasiya Chernenko and Sinikka Kollanus for technical assistance. We also thank the University of Helsinki core facilities Biomedicum Imaging Unit (BIU), Genome Biology Unit (supported by HiLIFE and the Faculty of Medicine) and the Biomedicum Imaging Core (BIC) at Karolinska Institutet. This work was funded by Sigrid Juselius Foundation (to K.L., O.C.), Finnish Cancer Foundation (to K.L., E.A.P., O.C.), Orion Research Foundation (to E.A.P.), K. Albin Johanssons Foundation (to K.L., E.A.P.), Emil Aaltonen Foundation (to E.A.P.), KI Strategic Research Program in Cancer (StratCan-KICancer; to K.L.), Swedish Cancer Society (2015/723, 2018/858 to K.L., J.C.), Swedish Research Council (2019-01541 to K.L.), the European Union's Horizon 2020 research and innovation program under grant agreement No 667403 for HERCULES (to S. Hautaniemi, O.C., S.G.), and the Doctoral Programme in Integrative Life Science (to E.A.P.).

## Author contributions

Conceptualization and methodology: E.A.P., J.G.-M. and K.L. Data acquisition, generation and analysis: E.A.P., J.G.-M., L.M.-G., S.J., K.Z, L.L., S.P.T., T.A.M., O.G., T.L., K.K., U.J., J.H., S. Hietanen, R.L, S. Hautaniemi, O.C., K.L. Clinical coordination: U.J., J.H., S. Hietanen, S.G. Writing, review, and/or revision: E.A.P., J.G.M., L.M.-G., K.L. Principal investigators: S.G., S. Hautaniemi, O.C., J.W.C., K.L. Supervision: K.L.

## Competing interests

The authors declare no competing interests.
