## [Peer Review File · Nature Communications]

REVIEWER COMMENTS

Reviewer #1 (Remarks to the Author): Expert in ovarian cancer resistance

Utilizing clinical samples and gene expression analysis followed by functional assays this manuscript evaluates the potential role of the extra-cellular matrix in modulating resistance to chemotherapy. Authors show that there is up regulation of certain collagens including Collagen IV (Col6) in tumors treated with chemotherapy. The initial analysis is performed using matched primary and metastatic tumor sites from patients treated with chemotherapy (some receiving neoadjuvant chemotherapy and others presumably treated with primary surgery then chemotherapy and had recurrent disease?). Following work is done using three cell lines Ovc4r-4 and Ovc4r 8 and T both known to be resistant to carboplatin and TKY-nu cell line known to be more sensitive to platinum drugs.

Understanding mechanisms that regulate therapy resistance in ovarian cancer has tremendous clinical value and the authors are congratulated on this study focused on the role of extracellular matrix and tissue microenvironment as it relates to therapy resistance.

I have the following questions

1) are the authors able to perform the same bioinformatics analyses segregating and comparing patients with carboplatin sensitive to those with carboplatin resistant disease? This would be very important as some patients who receive neoadjuvant chemotherapy may have a very good response to treatment and have platinum sensitive disease vs. others who may have platinum resistant/refractory disease. If this clinical information is available would be helpful to include it and also to perform analyses separately for patients with platinum resistant vs. sensitive disease looking at pre-chemotherapy and also post chemotherapy tumors.

2) When describing the clinical samples it is sometimes difficult to distinguish the following: a) Comparison of same site disease pre then post therapy (e.g. ovarian tumor or mental tumor) vs. b) comparison of metastatic disease site vs. primary tumor site (omental disease vs. ovarian/tubal disease). Can the manuscript be clarified so this is easier to follow.

3) What is the rationale for picking the proposed cell lines? Ovc4r-8 and Ovc4r-4 cells are both platinum resistant. Would it be helpful to include a platinum sensitive line such as Ovc4r-3 for comparison purposes?

4) Can the authors better describe the models used to assess stiffness: for example not clear what is meant by COL-1 functionalized soft and stiff gel- are these 2-D culture models.

5) Similarly it could be helpful to separate the impact of the matrix on survival vs. drug response- reading the manuscript this is sometimes not clear.

6) In experiments where primary patient samples are used it would be helpful to outline the number of samples tested

7) can the authors examine expression levels of Col6 in matched tumors longitudinally obtained from patients who started with platinum sensitive disease then developed platinum resistance?

8) Do the authors believe that this matrix particularly Col6 is secreted autonomously by tumor cells or by cells in the microenvironment? Is there a way to model the disease in vivo in the presence and absence of Col6 with and without chemotherapy to assess its functional role?

Reviewer #2 (Remarks to the Author): Expert in genomics

In this manuscript, the authors investigate the role of ECM components, in particular collagen 1, collagen 6 and fibronectin in ovarian high grade serous carcinoma and the response to platinum-based chemotherapy. They utilise established cell lines, in particular OVCAR4 and OVCAR8, primary cultures and primary samples from patients both before and after platinum-based chemotherapy. The work examines gene expression before and after chemotherapy from multiple anatomical sites, screens using individual ECM components and assessment of the impact of matrix stiffness. Overall the main conclusion is that specific components of the ECM can regulate platinum resistance.

The results are somewhat observational and there are several points where the authors' conclusions do not appear to be fully supported by the presented data – in particular, there is a tendency to equate association with causation.

However, overall, the work is technically extremely impressive, and the authors have utilised their primary material very well. The work is important and addresses a critical question in ovarian cancer.

General points

1. Primary samples used in Figure 1 and 2. As stated above, the authors make excellent use of these samples. However, if I understand correctly, the pre- and post-chemotherapy samples are not matched and thus derive from different patients. This poses a problem – patients who are selected by tumor boards/multi-disciplinary teams to undergo primary surgery generally have a better prognosis than those who are treated with primary chemotherapy (a very good example is shown in Clamp et al Lancet 2019, figure 2B). Thus, the pre- and post-chemotherapy samples used here will derive from prognostically different groups. This at least needs to be commented upon. However, also see point 6.

2. There is now a validated pathological chemotherapy response score (CRS) for ovarian cancer samples taken at interval debulking surgery following primary chemotherapy, which is strongly prognostic (see Böhm S JCO 2015 and Cohen et al Gynecol Oncol 2019) and has now been incorporated into reporting guidelines (see McCluggage et al Mod. Pathol. 2015). It would be most informative to correlate the changes in gene expression described here with CRS and patient outcomes – especially as fibro-inflammatory changes are incorporated into CRS.

3. The immune cell infiltration following chemotherapy in Fig 2h,i and Fig S2g,h – although H&E staining can be used to identify immune cells, further characterisation using IHC is recommended – what is the nature of these immune cells? In addition, the comment on lines 162 – 163 “The enhanced post-chemotherapy MMP activity can, therefore, derive from the therapy-induced leucocyte infiltration” is not really justified – what is the evidence that the leucocytes are driving the MMP activity rather than merely responding?

4. Stiffness vs resistance in Figure 3. These are very interesting data. Increasing stiffness in COL1-

functionalised gels increases cell replication (in absence of platinum) and reduces cisplatin-induced Casp3 cleavage. However, there is either an increase (OVCAR4) or no change (OVCAR8) in gH2Ax formation in response to platinum. This implies that there is failure of DNA damage response signalling with increasing stiffness, which may be critical to response to platinum-based chemotherapy. However, this is not explored further. Another example of association vs causation is found in lines 195 – 197: “the stiff TME in advanced omental metastases protects cancer cells from platinum-induced apoptosis in connection with changes in cell morphology and ECM adhesion” Again, I am not sure that the data presented fully justify that statement in relation to the omental samples.

5. Figure 4 is central to the paper as these experiments investigate direct effects of ECM components upon response to platinum. There are two points to discuss. Firstly, there appear to be no universal effects; COL1 and FN have opposite effects in OVCAR4 and OVCAR8 for example. Secondly, the results indicate that sensitivity correlates positively with cell counts – thus, there is a direct link between sensitivity and proliferation. However, it is not clear if the resistance is purely a result of proliferation. Would it be possible to repeat these experiments in low serum conditions or other conditions that will reduce proliferation to separate resistance from proliferation potentially?

6. Figure 5d-g is also very important. This demonstrates that the addition of COL6 to COL1 increases platinum resistance in OVCAR8 but not in OVCAR4. Similarly, in Figure 7, the COL1+COL6 combination increases resistance in primary cells from patients with recurrent disease but not chemotherapy-naïve patients. This implies that COL6 acts upon resistance mechanisms that are already active, given that OVCAR8 cells are intrinsically less sensitive to platinum than OVCAR4. This raises the key question of tumour cell autonomous changes that drive interactions with ECM – OVCAR8 have, in addition to TP53 mutation, mutations in ERBB2, KRAS (although not classic G12D/C mutation) and CTNNB1 – have the authors undertaken any genomic analyses of their r-HGSC cells to correlated with OVCAR8?

7. Finally, what is the effect of inhibition beta1 integrin on platinum resistance in these models? If the authors are correct, beta1 integrin blockade in r-HGSC in the presence of COL6 should reverse the resistance phenotype.

Reviewer #3 (Remarks to the Author): Expert in ECM/stiffness

This manuscript uses RNA-seq to identify the matrisome as differential expressed in primary tumors compared to metastatic sites. This finding is important and novel. However, the remainder of the paper is hard to follow and somewhat disconnected. Conclusions are overstated in places. There are many different assays run without a clear connection in logic between them. Much of the data appears to be highly correlative. Overall, tases are important findings but the construction of the manuscript needs improvement. Some more specific comments include:

-The authors state: The enhanced post-chemotherapy MMP activity can, therefore, derive from the therapy-induced leucocyte infiltration, leading to increased ECM degradation and remodeling, undetectable by the matrisome transcription. However, they have not shown causation here. The language should be corrected so as to not sounds like a conclusion.

-The manuscript jumps around a bit, making it hard to follow. As one example, the transition from the sequencing data to the cell culture experiments is unclear. Also, the connection to the MMP DEG signature is not clear.

-The results using PA gels in figure 3 are expected based on the past 20 years of using these substrates. Most groups now are focused on 3D matrices since there are known differences between 2D and 3D cell response.

-The reason that cells responded differently (figure 3G) is not clear, and as such it is not clear how the data should be interpreted.

-Much of the data is correlative. As an example, the data on COL6 (Figure 5g-h) is not demonstrated definitively as a protector for chemoresistance. Much of the data and interpretation is based on comparison of 2 cells lines OVCAR4 and OVCAR8 which is not sufficient to draw a conclusion.

We thank the Reviewers for the interest in our work and appreciate the constructive comments, which motivated us to conduct several new analyses and allowed important improvements to the manuscript as described below. As requested by the editor, we particularly addressed the concerns of the Reviewers with regards to the presentation of the clinical samples, provided further evidence of causality of ECM stiffness and additionally included the platinum sensitive OVCAR3 cell line in new experiments.

While performing the requested new analyses of our transcriptomic data and carefully annotating patient and sample information as suggested by the Reviewers, we identified four mislabeled samples and rigorously repeated all the affected analyses in the original manuscript. The new results are included in new Fig. 1, 2, 5, Supplementary Fig 1, 2, 3, 10, 13 and corresponding Supplementary Data.

Reviewer #1: Expert in ovarian cancer resistance

1) Are the authors able to perform the same bioinformatics analyses segregating and comparing patients with carboplatin sensitive to those with carboplatin resistant disease? This would be very important as some patients who receive neoadjuvant chemotherapy may have a very good response to treatment and have platinum sensitive disease vs. others who may have platinum resistant/refractory disease. If this clinical information is available would be helpful to include it and also to perform analyses separately for patients with platinum resistant vs. sensitive disease looking at pre-chemotherapy and also post chemotherapy tumors.

Response: This is an important suggestion for improving our study. In the original manuscript, we had investigated global gene expression changes in the matrisome from different disease sites (primary ovary/fallopian tube tumors, metastatic omental, peritoneal and mesenteric tissues, and ascites) pre- and post-chemotherapy. As suggested, we have now also analyzed the matrisome gene expression in patients segregated based on their platinum free interval (PFI) into the groups of platinum resistant disease (PFI \leq 6 months) or platinum sensitive disease (PFI $>$ 6 months), both pre- and post-chemotherapy in primary tumor and metastatic tissues (combination of omentum, peritoneum and mesentery; see new Supplementary Fig. 3a for the distribution and number of samples in each analysis).

In primary tumor samples from platinum resistant versus sensitive patients, we identified 22 differentially expressed genes (DEGs) pre-chemotherapy. The majority of these genes encoded ECM glycoproteins and matrisome-associated proteins upregulated in patients with resistant disease (new Supplementary Fig. 3b). Post-chemotherapy, we identified 32 DEGs in primary tumor samples, with a substantial proportion increased in resistant patients and consisting of collagen encoding genes, including *COL11A1* (new Supplementary Fig. 3c), which has been previously described as part of an ECM-gene signature that associates with the extent of disease and tissue stiffness in treatment-naïve HGSC omental metastasis¹.

In similar analyses of combined metastatic tissues, we found 73 DEGs pre-chemotherapy (new Supplementary Fig. 3d). Unexpectedly, the majority of these DEGs were downregulated in patients with platinum resistant disease, including genes encoding collagens and secreted factors, such as *COL11A2* and *IL6*, while only 11 genes including chemokines were upregulated. In the combined metastatic tissues post-chemotherapy, we found total of 15 DEGs among which *CCL28* was the most significantly upregulated and *ITNL1* the most downregulated matrisome gene in the resistant disease (new Supplementary Fig. 3e).

Our originally submitted results showed that during chemotherapy both primary and metastatic tissues undergo major changes on the matrisome (see Fig. 2b,c in the revised manuscript), yet the

core matrisome proteins, including several collagens, remain more abundant in the metastatic tissues compared to the primary tumors (see Fig. 2e). When segregating the patients based on their PFI, we observed that the primary tumors of resistant patients were enriched for core matrisome proteins post-chemo (new Supplementary Fig. 3c), whereas minor differences of the core matrisome were observed in the metastatic tissues between sensitive and resistant patients post-chemo (new Supplementary Fig. 3e). These results are consistent with a concept that chemo-induced changes in the already desmoplastic metastatic sites do not differ drastically between platinum resistant versus sensitive patients or pre- versus post-chemo, whereas the primary tumor in resistant as well as post-chemo patient tumors becomes more collagenous/fibrotic.

These additional results have now been included in the revised manuscript Results (pages 7-8) and in a new Supplementary Fig. 3.

We also performed the same analysis for ascites-derived cells from patients with platinum sensitive versus platinum resistant disease, both pre- and post-chemotherapy (excluding samples collected at relapse-stage due to majority belonging to patients with short PFI). This resulted in very low sample number, particularly for the post-chemo comparison (n = 2 resistant versus n = 1 sensitive patient ascites). Thus, we have not included the results in the new Supplementary Fig. 3. However, also this data has been included in Supplementary Data 24, 25. Although with too small n to draw solid conclusions, it is of potential interest regarding the co-evolution of cancer cells and their microenvironment that collagen encoding genes including *COL4A1-2*, *COL5A1*, *COL6A1* and *COL26A1* were upregulated in the two resistant patients' ascites cells relative to those from the single patient with sensitive disease post-chemo. This notion is also relevant regarding to the comment 8 from this Reviewer.

2) When describing the clinical samples it is sometimes difficult to distinguish the following: a) Comparison of same site disease pre then post therapy (e.g. ovarian tumor or mental tumor) vs. b) comparison of metastatic disease site vs. primary tumor site (omental disease vs. ovarian/tubal disease). Can the manuscript be clarified so this is easier to follow.

Response: We thank the Reviewer for this constructive comment. In order to clarify the presentation, we have now modified Results on pages 4-8 and corresponding figures to clearly indicate whether the analyses compared different tumor sites or matching tissue types pre- and post-chemotherapy. We have also revised Methods pages 24-25 to clarify the patient material used in this study.

The sample types used in this manuscript include ovary/fallopian tube tumors (referred as primary tumors), metastatic tumors from omentum, peritoneum and mesentery (referred as metastatic tissues when grouped for the analyses) and ascites-derived cancer cells. Samples were obtained at three different timepoints: pre-chemotherapy (diagnostic laparoscopy for patients subjected to neoadjuvant chemotherapy treatment/NACT or primary debulking surgery/PDS), post-chemotherapy (interval debulking surgery) and at post-chemotherapy relapse-stage from both NACT and PDS patients (see Supplementary Data 1 for detailed identification of sample collection stage).

In Results, we describe gene expression comparison between the metastatic and primary sites pre-chemotherapy (Fig. 1, Supplementary Fig. 1) as well as between pre- and post-chemotherapy on each site (Fig. 2a-d, Supplementary Fig. 2a-c in the revised manuscript). In addition, we show the comparison of matrisome expression between post-chemo omental+peritoneal metastatic tissues and primary sites. To clarify Results text in this regard, we have now re-organized the results in Fig. 2 to first describe all the pre- versus post-chemo comparison results (Fig. 2a-d, Supplementary Fig. 2a-c) and only then bring up the analysis results between different post-chemo samples (Fig. 2e-f in the revised manuscript). To provide as complete and clear data as possible, we have now also included the comparison of omental+peritoneal metastases pre- versus post-chemotherapy (see

new Fig. 2c and Supplementary Data 15). In the original manuscript, data on this step was not shown, although comparison was included for the omental+peritoneal metastases versus primary tumors post-chemo.

To address the first comment from Reviewer#2, we have now also included a detailed patient information per sample in Supplementary Data 1 and 2 in order to clarify the distribution of samples obtained either from patients undergoing NACT or PDS (see also comment 1 from Reviewer#2).

3) What is the rationale for picking the proposed cell lines? Ovar-8 and Ovar-4 cells are both platinum resistant. Would it be helpful to include a platinum sensitive line such as Ovar-3 for comparison purposes?

Response: The selection of cell lines to study high-grade serous ovarian carcinoma (HGSC) is an important question to consider. Indeed, recent findings have brought into light that up to 90% of published studies using “HGSC” cell lines are actually based on cells that do not recapitulate the mutational landscape of HGSC². In this study we had initially used four cell lines: OVCAR4, OVCAR8, TYK-nu parental and the experimentally produced more resistant subline TYK-nu.R³. All these cells recapitulate the mutational landscape of HGSC⁴, and were selected based on their varied platinum sensitivity *in vitro*, described by us and others^{5,6}. Briefly, in our experiments we have found OVCAR4 and TYK-nu cells to be relatively more platinum sensitive *in vitro* in comparison to OVCAR8 and TYK-nu.R, respectively (see new Supplementary Fig. 6a as well as Moyano-Galceran *et al*⁶ for the 2D cisplatin sensitivity comparison of these cells). The cell line couple of TYK-nu and TYK-nu.R was included based on their uniform genetic background to consider the relationship between the cancer cell autonomous and ECM-dependent generation of platinum resistance.

As suggested by the Reviewer, we have now further repeated some of the key experiments of our study using OVCAR3 and thus validated our previous results regarding the COL6-mediated protection against treatment selectively in the platinum resistant cells (see new Fig. 6f-h, where in contrast to the resistant TYK-nu.R, cell viability was not increased upon treatment in the relative platinum sensitive OVCAR3 and TYK-nu on COL6). During the revision, we also used OVCAR3 to investigate the link between ECM-dependent proliferation and platinum sensitivity suggested by Reviewer#2 (comment 5), which results have now been included in the new Supplementary Fig. 8i,j (see also response to this comment 5 to Reviewer#2).

4) Can the authors better describe the models used to assess stiffness: for example not clear what is meant by COL-1 functionalized soft and stiff gel- are these 2-D culture models.

Response: To clarify the presentation of these models, we have now revised the text in Results page 9 to following: “we used soft (2kPa) and stiff (21 kPa) polyacrylamide hydrogels functionalized for cell adhesion with covalently-bound COL1”. The exact details of the gel preparation had been included also in the originally submitted manuscript and can now be found in Methods pages 27-28 in the revised manuscript.

5) Similarly it could be helpful to separate the impact of the matrix on survival vs. drug response-reading the manuscript this is sometimes not clear.

Response: This is a valid point. In this study we have used different endpoint measurements to investigate the effect of ECM on cell survival and cisplatin response, where latter was determined by the change in cell count or viability upon treatment (e.g. (cell count [NaCl]– cell count [cisplatin]) / cell count [NaCl]). In order to study matrix-mediated survival, we have determined cell apoptosis by cleaved caspase 3 (original Fig. 3, 6 and Supplementary Fig. 3, 7) and cell viability by cell count (original Fig. 4a,b) as well as by ATP-based cell activity (original Fig. 7f-h and Supplementary Fig.

5c,d). To determine ECM-mediated cisplatin response, we have used cell count (original Fig. 4c) as well as ATP-based measurements of cell viability (original Fig. 4d).

We admit that the use of cell count and ATP-based measurements as an endpoint for both cell survival and cisplatin response may have led to confusion. In addition, we realized that the concept of “drug response” (cisplatin response, platinum response, chemoresponse, treatment response) in the context of ECM effects had been used in a variable manner in different parts of the manuscript. To address these issues, we have now defined the concept of cisplatin response in page 11 as “change in cell count or cell viability upon treatment”, and used it accordingly in the description of Fig. 4 and Supplementary Fig. 8.

In order to clarify the use of different concepts, and to also address the changes requested by Reviewer#2 in comment 4 and Reviewer#3 in comment 2, lines 164-166, 173-175, 365-366 from the original manuscript have now been reformatted. As an example, lines 365-366 in the original manuscript: “In this study, we present comprehensive expression signatures of longitudinal HGSC cohort along with in-depth **platinum response** results with cell models...”, has now been changed as follows: “In this study, we present comprehensive expression signatures of longitudinal HGSC cohort along with in-depth results of **cellular responses to platinum** investigated in cell models...”

6) In experiments where primary patient samples are used it would be helpful to outline the number of samples tested

Response: In our manuscript we have used primary patient samples in Fig. 1, 2, 5, 8 and Supplementary Fig. 1-5, 10, 13 (original Fig. 1, 2, 5, 7 and Supplementary Fig. 1, 2, 6, 7). In the originally submitted manuscript the number of samples included in each experiment and analyses was described either in the Results (for transcriptomic data and immunohistochemistry in Fig. 1, 2 and Supplementary Fig. 1, 2), in the figure (transcriptomic data in Fig. 1a,b, 2a, 5b and Supplementary Fig. 6b) or in the figure and figure legend (Fig. 7 and Supplementary Fig. 7c,d).

To improve the consistency of the description of the number of samples in each step, we have now added the number of tested primary patient samples both in the Results and legend to each figure, also to those figures where n has been described in the figure (e.g. Fig. 5b in revised manuscript). For Fig. 8 (original Fig. 7) we have emphasized that each graph represents one individual HGSC patient-derived organoid culture, i.e. in Fig. 8g. n = 3, where n refers to replicates of one individual patient cell cultures.

7) Can the authors examine expression levels of Col6 in matched tumors longitudinally obtained from patients who started with platinum sensitive disease then developed platinum resistance?

Response: We thank the Reviewer for this excellent suggestion. To study whether the expression of COL6 is altered in those patients who despite initial chemotherapy sensitivity develop platinum resistant disease, we have now analyzed COL6 (*COL6A1*, *COL6A2*, *COL6A3*, *COL6A5* and *COL6A6*) expression in matched pre- and post-chemotherapy samples (n = 13; new Fig. 5d and Supplementary Fig. 10c). By using samples collected from primary tumor and metastatic tissues, we show that in these patients who developed chemotherapy resistance, the expression of *COL6A1-6A3* was increased during the disease progression and/or chemotherapy treatment. Despite the overall lower expression in comparison to the solid tissues, the expression of *COL6A1-6A3* in ascites-derived cancer cells was also increased. The expression of *COL6A5-6A6*, encoding the alternative polypeptides to *COL6A3* in the trimeric COL6 protein⁷, was more variably affected.

An exception to the increased COL6 expression from pre-chemo to post-chemo was patient EOC218 (marked with an asterisk in Fig. 5d and Supplementary Fig. 10c), where the expression of *COL6A1-6A3* decreased over time. Of note, the PFI of this patient was 974 days, whereas the PFI

of other patients varied between 0-460 days (see Supplementary Data 30 for corresponding gene expression, fold change, PFI and PFS in all matched pre-post RNA sequenced samples included in Fig. 5c,d and Supplementary Fig. 10c).

As suggested by Reviewer#2, we also performed PFI and PFS analyses against COL6 expression in solid tumor tissues including all matched pre-post chemo patient samples (n = 13, which distinct to the comparison described above, also includes samples from progressive disease after NACT but excludes ascites samples; new Fig. 5c). These results further validate the correlation of COL6 expression in the solid tissues with both decreased PFI and PFS, highlighting its potential as a marker for poor survival and platinum response. These results have now been included in the new Fig. 5c and Supplementary Data 30. See also comment 2 to Reviewer#2.

8) Do the authors believe that this matrix particularly Col6 is secreted autonomously by tumor cells or by cells in the microenvironment? Is there a way to model the disease *in vivo* in the presence and absence of Col6 with and without chemotherapy to assess its functional role?

Response: This is a very interesting and relevant question. Previously overexpression of COL6A3 by ovarian cancer cell lines has been linked to *in vitro* cisplatin-resistance, whereby the cancer cell-secreted COL6 contributed to the increased resistance⁸. In our originally submitted manuscript, we showed by transcriptomic data that expression of COL6 genes is much higher in the stroma-containing tumor tissues than in the ascites-derived cancer cells with less ECM-producing stroma (Fig. 5b and Supplementary Fig. 10b in the revised manuscript). Moreover, immunohistochemistry of COL6A1 in omental micro-metastasis as well as in pre-chemo and post-chemo omental metastases, show strongest COL6 protein accumulation in areas of stroma surrounding the cancer colonies (original Fig. 5c, see Fig. 5g in the revised manuscript).

To address this question, we further analyzed a publicly available dataset of HGSC patient samples⁹. The expression of each COL6 gene was markedly higher in the stromal compartment than in the epithelial compartment (see new Fig. 5e). We also assessed COL6A1 protein in HGSC cell lines, patient ascites-derived cells, patient-derived cancer-associated fibroblasts (CAFs), and normal fibroblasts (NFs). Albeit with somewhat unequal loading, the results indicate that the analyzed CAFs and NFs expressed COL6A1, but it was undetectable or very low in cancer cells (both cell lines and patient ascites-derived cells; see new Fig. 5f). HGSC patient-derived ascites cells used in this experiment were obtained from patients undergoing PDS at Karolinska University Hospital. This patient sample collection, culture and characterization has been done in collaboration with the responsible surgeon Dr Joneborg, who thus has been added as a new co-author in the revised manuscript.

In the matched pre- and post-chemo patient samples COL6A1-6A6 gene expression was also notably stronger in the solid tumor tissues than in ascites-cancer cells (matched samples from platinum sensitive patients who developed resistance to treatment; see above comment 7). Yet, the expression was increased in both solid tumors and ascites cells upon the disease progression. The very preliminary comparison of post-chemotherapy ascites-derived cancer cells between platinum resistant (n = 2) and sensitive (n = 1) disease, is also consistent with the idea that, in addition to the stroma, the cancer cells can express COL6 particularly upon the development of resistant disease (new Supplementary Data 25). However, these results need further validation.

Regarding the question of *in vivo* modeling of the disease in COL6-abundant versus depleted microenvironments to study the functional role of this ECM, we are aware of a COL6 deficient mouse model developed by Iyengar et al¹⁰, which has been used to study the effect of adipocyte-derived COL6 in breast cancer.¹⁰ Considering the strong stromal COL6 expression that we revealed in human HGSC, a relevant transgenic tumor model or use of COL6+/- mouse cancer cells in COL6-deficient background could be used to assess the functional role of COL6.

Reviewer #2 Expert in genomics

1) Primary samples used in Figure 1 and 2. As stated above, the authors make excellent use of these samples. However, if I understand correctly, the pre- and post-chemotherapy samples are not matched and thus derive from different patients. This poses a problem – patients who are selected by tumor boards/multi-disciplinary teams to undergo primary surgery generally have a better prognosis than those who are treated with primary chemotherapy (a very good example is shown in Clamp et al Lancet 2019, figure 2B). Thus, the pre- and post-chemotherapy samples used here will derive from prognostically different groups. This at least needs to be commented upon. However, also see point 6.

Response: As the Reviewer points out, the pre- and post-chemotherapy samples in our cohort are not exclusively from matched patients (matching pre- and post-chemotherapy samples $n = 19$; matching samples included in RNAseq $n = 16$), and that patients have undergone either NACT (71% of the cohort) or PDS (29%) treatment strategy. Highlighting the importance of Reviewers comment, and the findings by Clamp et al¹¹, NACT patients in our cohort had shorter PFI in comparison to PDS patients ($p = 0.008$, log rank). Therefore, there is indeed some asymmetry in our cohort. The summary of the sample distribution and corresponding PFS / PFI based on the treatment-arm have now been added to Supplementary Data 2.

Majority of the PDS samples in our study were collected pre-chemotherapy (46/49; see Table 1 for Reviewers for the sample distribution between treatment-arms). As much as 75% of the primary tumors were from patients subjected to PDS, whereas 38% of omental, 21% of peritoneal and 17% of mesenteric metastases samples consisted of PDS samples (Table 1 for Reviewers). We thus considered that in our gene expression analysis, this asymmetry could primarily affect the pre-post-chemotherapy comparison, particularly for primary tumors (in Fig. 2 and Supplementary Fig. 2).

In order to estimate the effect of such asymmetry between PDS and NACT patients, we analyzed DEGs using only NACT patient samples. Of these, 51-93% were shared with the DEGs identified in the combined NACT + PDS samples, and these portions did not fully correspond to those of the PDS samples. Therefore, we think that, although the patients undergoing different treatment-arms represent prognostically different groups, there is also the heterogeneity within each tissue site, and thus the use of a larger sample set instead of segregating patients based on the treatment scheme can be a more relevant strategy for our expression analysis.

To further address this imbalance between PDS and NACT patients' PFI, and to also investigate whether this would have an effect in our functional studies and findings of COL6-induced chemoresistance in relapse HGSC (r-HGSC) patient-derived ascites cancer cells (Fig. 8 in revised manuscript /original Fig. 7) we also checked the treatment-arm of the used pre-chemo (p-HGSC) and r-HGSC samples. Suitably, these samples were obtained from both patients undergoing NACT (3/7; EOC1192/p-HGSC, EOC198/p-HGSC, EOC677/r-HGSC) and PDS (4/7; EOC1120/p- and r-HGSC, EOC50/p-HGSC, EOC495/r-HGSC, EOC742/r-HGSC). Therefore, these data support our findings of ECM-induced intrinsic or acquired resistance in r-HGSC, despite of the prognostic grouping of PDS / NACT treatment-arms. This information has now also been included in Results

Table 1 for Reviewers. Sample distribution.

Treatment strategy	NACT		PDS	
	Pre-chemo, n	Post-chemo, n	Pre-chemo, n	Post-chemo, n
Primary tumor (RNAseq)	10 (8)	14 (13)	24 (24)	
Omental metastasis (RNAseq)	17 (13)	29 (23)	8 (8)	
Peritoneal metastasis (RNAseq)	22 (22)	5 (5)	6 (6)	
Mesenteric metastasis (RNAseq)	5 (5)	10 (9)	1 (1)	(1)
Ascites (RNAseq)	8 (4)	16 (16)	7 (5)	4 (2)

NACT = Neoadjuvant chemotherapy treatment; PDS = primary debulking surgery; RNAseq = number of samples in transcriptomic dataset.

2) There is now a validated pathological chemotherapy response score (CRS) for ovarian cancer samples taken at interval debulking surgery following primary chemotherapy, which is strongly prognostic (see Böhm S JCO 2015 and Cohen et al Gynecol Oncol 2019) and has now been incorporated into reporting guidelines (see McCluggage et al Mod. Pathol. 2015). It would be most informative to correlate the changes in gene expression described here with CRS and patient outcomes – especially as fibro-inflammatory changes are incorporated into CRS.

Response: We thank the Reviewer for this very good suggestion. The chemotherapy response score (CRS) is based on the examination of the omentum to assess the extent of tumor regression after NACT and it has been indeed associated to poor overall survival and PFS in ovarian cancer patients. However, this is not yet a standardized analysis/procedure and has not been incorporated in the pathological reports at the Turku University Central Hospital (TUCH, Finland) for the cohort used in this study. Moreover, a recent multicenter study has shown that the CRS is robustly predictive only with big cohorts¹². In our cohort, only 14 omental samples were obtained at interval debulking surgery after NACT.

However, to improve our manuscript, we investigated the change in the expression of the key genes in this study (*COL1*, *COL6*, *FN* and *VTM*) in all matched NACT patient-derived pre-post-chemotherapy solid tissues (n = 13; matching primary tumor and metastatic tissues) and correlated the change with PFI / PFS. These results revealed that the upregulation of *COL6A1* and *COL6A2* in these bulk tumor tissues upon chemotherapy and disease progression significantly correlated with shorter PFI (*COL6A1*: r = -0.77; p = 0.002; *COL6A2*: r = -0.73; p = 0.005; Pearson correlation) and PFS (*COL6A1*: r = -0.76; p = 0.002; *COL6A2*: r = -0.71; p = 0.007; Pearson correlation). Despite we had no possibility to assess the CRS in these samples, these results suggest that the *COL6* upregulation in solid HGSC tissues could provide prognostic information to evaluate treatment outcome after chemotherapy. However, in order to speculate whether this change is specific for omental metastases or all solid HGSC tissues, and how the change correlates with CRS, a larger patient cohort should be studied. These results are now included in the new Fig. 5c and Supplementary Data 30.

3) The immune cell infiltration following chemotherapy in Fig 2h,i and Fig S2g,h – although H&E staining can be used to identify immune cells, further characterisation using IHC is recommended – what is the nature of these immune cells? In addition, the comment on lines 162 – 163 “The enhanced post-chemotherapy MMP activity can, therefore, derive from the therapy-induced

leucocyte infiltration” is not really justified – what is the evidence that the leucocytes are driving the MMP activity rather than merely responding?

Response: We thank the Reviewer for the constructive suggestion to characterize the immune cell infiltration in the post-chemotherapy metastatic HGSC tissues. We also agree that in our original manuscript the conclusions regarding the link between MMP activity and leukocyte infiltration were unjustified, although based on literature¹³. We have now revised the manuscript and modified the text accordingly, discussing the link between MMP activity and immune cell infiltration in the Discussion page 20.

While we think that a comprehensive analysis of the nature of the immune cells in these varying tumor tissues and disease stages is beyond the scope of this study, we have now performed immunohistochemistry for CD45 (leukocytes), CD68 (macrophages), CD79a (plasma cells) and CD3 (T-cells) in the post-chemotherapy mesentery and omental metastatic tissues. The results revealed leukocyte infiltration in these post-chemotherapy tissues (see new Fig. 2k-l and Supplementary Fig. 5), suggestive of tumor-associated inflammation and leukocyte infiltration, induced by chemotherapy and/or disease progression.

4) Stiffness vs resistance in Figure 3. These are very interesting data. Increasing stiffness in COL1-functionalised gels increases cell replication (in absence of platinum) and reduces cisplatin-induced Casp3 cleavage. However, there is either an increase (OVCAR4) or no change (OVCAR8) in γ H2Ax formation in response to platinum. This implies that there is failure of DNA damage response signalling with increasing stiffness, which may be critical to response to platinum-based chemotherapy. However, this is not explored further. Another example of association vs causation is found in lines 195 – 197: “the stiff TME in advanced omental metastases protects cancer cells from platinum-induced apoptosis in connection with changes in cell morphology and ECM adhesion” Again, I am not sure that the data presented fully justify that statement in relation to the omental samples.

Response: We agree with Reviewer#2 that understanding putative differences in DNA damage response signaling and repair mechanisms in HGSC cells at different COL1 stiffness is of interest. However, we think that a comprehensive investigation of such mechanisms is out of the scope of our study. Previously, the methylation of *BRCA1* in OVCAR8¹⁴ has been reported and it is known that cells lacking *BRCA1* depend on *chk1* for the repair of endogenous DNA damage¹⁵. Thus, to understand whether the significantly lower γ H2Ax foci intensity in OVCAR8 at high stiffness (Fig. 3f) could be explained by differences in DNA damage response signaling, we inhibited *chk1/2*, downstream of the HA2X phosphorylating ATM/ATR-complexes. As expected, *chk1/2* inhibition resulted in sensitization of HGSC cells to cisplatin, specifically in stiff substrates (Fig. 3l)¹⁶. However, OVCAR8 but not OVCAR4 showed more platinum-induced DNA damage upon *chk1/2* inhibition (Fig. 3m), suggesting that *chk1/2* mediate protection against DNA damage in OVCAR8.

Regarding the statement in lines 195-197, we admit that it was an overstatement and have now modified the text to: “Altogether, these results suggest that evasion of cisplatin-induced apoptosis via DNA damage response & repair pathways and stiffness-dependent adhesion signaling can occur via distinct, overlapping mechanisms”.

5) Figure 4 is central to the paper as these experiments investigate direct effects of ECM components upon response to platinum. There are two points to discuss. Firstly, there appear to be no universal effects; COL1 and FN have opposite effects in OVCAR4 and OVCAR8 for example. Secondly, the results indicate that sensitivity correlates positively with cell counts – thus, there is a direct link between sensitivity and proliferation. However, it is not clear if the resistance is purely a result of proliferation. Would it be possible to repeat these experiments in low serum conditions or

other conditions that will reduce proliferation to separate resistance from proliferation potentially?

Response: We thank the Reviewer for this important consideration between proliferation and cellular response to platinum.

Regarding the first point, we agree that none of the studied ECMs resulted in equal cellular response to platinum in all the cell lines used in the original manuscript. However, these cells are different at the phenotypic level (epithelial versus mesenchymal)⁵, have different platinum sensitivities (see new Supplementary Fig. 6a) and even differ in their genetic background (see response to next comment 6). Albeit these differences, the glycoproteins fibronectin (FN) and vitronectin (VT) promoted chemoresistance in OVCAR4, TYK-nu and TYK-nu.R, whereas OVCAR8 were more resistant to platinum in collagen-based substrates (see Supplementary Fig. 8a,b,e,f in revised manuscript).

To address the second point, we investigated the effects of these ECMs (FN, VTN, COL1 and COL6) on platinum resistance in low serum conditions (reduction from 10% to 1%). These experiments were performed using the 4 original cell lines and adding OVCAR3, following the recommendation of Reviewer#1 (comment 3). The growth of OVCAR3 and OVCAR4 on these substrates was similar at 10% and 1% serum, whereas OVCAR8, TYK-nu and TYK-nu.R grew significantly less in all substrates at 1% serum (see new Supplementary Fig. 8j). In this condition of reduced proliferation, cisplatin response of OVCAR3, OVCAR4, and TYK-nu (determined by the number of adherent cells) varied in an ECM-dependent manner but did not fully correlate with the corresponding ECM-dependent growth rates (see new Supplementary Fig. 8j). In contrast, OVCAR8 and TYK-nu.R cisplatin responses were instead diminished coincident with suppressed growth in low serum. Altogether, these results suggest that proliferation, variably affected by the ECM components is a factor, but not solely responsible for the prominent and variable ECM-dependent changes in cellular responses to cisplatin.

6) Figure 5d-g is also very important. This demonstrates that the addition of COL6 to COL1 increases platinum resistance in OVCAR8 but not in OVCAR4. Similarly, in Figure 7, the COL1+COL6 combination increases resistance in primary cells from patients with recurrent disease but not chemotherapy-naïve patients. This implies that COL6 acts upon resistance mechanisms that are already active, given that OVCAR8 cells are intrinsically less sensitive to platinum than OVCAR4. This raises the key question of tumour cell autonomous changes that drive interactions with ECM – OVCAR8 have, in addition to TP53 mutation, mutations in ERBB2, KRAS (although not classic G12D/C mutation) and CTNNB1 – have the authors undertaken any genomic analyses of their r-HGSC cells to correlated with OVCAR8?

Response: This is a very interesting suggestion. In order to start understanding whether the COL6-mediated increased chemoresistance observed in OVCAR8 and HGSC patient-derived cancer cells from relapse disease (r-HGSC) could be driven by additional mutations not found in the sensitive cell lines, we looked for ERBB2, KRAS and CTNNB1 mutations in the whole genome sequencing data of the r-HGSC cells. The mutational status of ERBB2 and KRAS is variable within the r-HGSC samples and none of the HGSC ascites cancer cells had CTNNB1 mutation (see Fig. 1 for Reviewers). Two out of four (one NACT patient-derived, one PDS patient-derived) r-HGSC patient-derived ascites cells used in this study (Fig. 8 in revised manuscript) harbored an ERBB2 mutation, while the other two r-HGSC patient-derived cells (both from PDS treatment-arm) had no mutation in ERBB2 or KRAS (Fig. 1 for Reviewers), suggesting that the intrinsic chemoresistance of r-HGSC-derived patients could be explained by other mutations and/or epigenetic factors. Such investigations will be of interest, but are considered above the scope of this study.

7) Finally, what is the effect of inhibition beta1 integrin on platinum resistance in these models? If the authors are correct, beta1 integrin blockade in r-HGSC in the presence of COL6 should reverse the resistance phenotype.

Response: This is a valid question. However, since different integrin heterodimers containing a $\beta 1$ subunit act as receptors for numerous ECM ligands, including COL1^{17,18}, we expected that blocking this integrin subunit would not solely have an effect on COL6 substrates. Our unpublished results indicate that integrin $\beta 1$ is essential for cancer growth and invasion in our 3D collagen models (see Fig. 2a for Reviewers)¹⁹. Using OVCAR4, OVCAR8, TYK-nu and TYK-nu.R cells in 3D COL1, we can observe effective blockage of cell growth and invasion after silencing integrin $\beta 1$ via transient siRNA transfection (silTG $\beta 1$; see Fig. 2a,b for Reviewers). However, cell viability was not affected by silTG $\beta 1$ upon cisplatin treatment (see Fig. 2b for Reviewers). Silencing ITG $\beta 1$ did not increase, but rather decreased apoptosis, as measured by caspase 3/7 activity (see Fig. 2c for Reviewers). These results suggest that silencing ITG $\beta 1$ would instead be mainly related to the reduced proliferation and invasion in 3D collagen-based models. Therefore, we did not further investigate the effects of blocking integrin $\beta 1$ in r-HGSC organoids grown in COL1 or COL1+COL6 matrices.

Reviewer #3 Expert in ECM/stiffness

1) The authors state: The enhanced post-chemotherapy MMP activity can, therefore, derive from the therapy-induced leucocyte infiltration, leading to increased ECM degradation and remodeling, undetectable by the matrisome transcription. However, they have not shown causation here. The language should be corrected so as to not sound like a conclusion.

Response: We agree with both Reviewer #2 and #3 that in our original manuscript the statement linking MMP activity and leukocyte infiltration could not be justified by our results, but was rather influenced by previous studies of MMP expression in leukocytes (reviewed in¹³). We have now modified the original manuscript to better report the findings of enhanced MMP activity post-chemotherapy independently of the leukocyte infiltration in Results (see pages 7-8). We have now also discussed this topic in the Discussion, as follows: "... the collagenous ECM appeared fragmented in conjunction with both increased protease-related transcription, predictive by IPA analysis of the ECM-degrading MMP activity, and leukocyte infiltration in the same metastatic tissues."

2) The manuscript jumps around a bit, making it hard to follow. As one example, the transition from the sequencing data to the cell culture experiments is unclear. Also, the connection to the MMP DEG signature is not clear.

Response: To improve the transition from the transcriptomic analyses we have now rewritten the conclusion of this part, highlighting the identified changes in core ECM upon disease progression and chemotherapy, which encompass both extent and composition. In this way we introduce the two concepts that will be investigated in the following sections of the manuscript: the extent of collagenous ECM, which dictates tumor stiffness and regulates biomechanical signaling (Fig. 3 and Supplementary Fig. 6), and the composition of the ECM, which determines substrate availability and controls adhesion signaling (Fig. 4 and Supplementary Fig. 7-9). See the modified text: "Altogether these data demonstrate that the fibro-inflammatory TME, closely surrounding the cancer-foci, changes markedly in response to disease progression and chemotherapy, including alterations on both the extent and content of the cancer-adjacent core ECM."

We have also rewritten the beginning of the next Results section emphasizing these two concepts, as follows: "The evolving matrisome can alter cancer cell functions in the fibrotic TMEs at least by 1) biomechanical signaling depending on the extent of collagenous ECM and consequent tumor stiffness, and 2) adhesion signaling based on the contents of specific ECM substrates."

We also acknowledge that the paragraph referring to the Ingenuity Pathway Analysis (IPA) in post-chemotherapy samples (original manuscript lines 146-152) was not properly linked to the findings in the previous paragraph, but rather referred to the IPA analysis in the previous section (pre-chemo samples in Fig. 1i). Following the recommendation to clarify the Results text (also commented by Reviewer#1), we moved the comparison of matrisome expression between post-chemo omental+peritoneal metastatic tissues and primary sites (Fig. 2e in revised manuscript) to precede the IPA analysis of these post-chemo samples (Fig. 2f in revised manuscript). After enumerating the most significant pathways, we then bring the result on MMP activity and compare it to the result from pre-chemo samples, as follows: "Opposite to the comparison between pre-chemo primary and metastatic tumors, however, the pathway activity for "Inhibition of MMPs" was reduced upon higher MMP expression in post-chemo omental and peritoneal metastases compared to the post-chemo primary tumors".

3) The results using PA gels in figure 3 are expected based on the past 20 years of using these substrates. Most groups now are focused on 3D matrices since there are known differences between 2D and 3D cell response.

Response: The Reviewer is right. In our manuscript we have used this relatively robust 2D model to solely investigate the effect of adhesion in HGSC cells, altering the biomechanical signaling induced by different substrate stiffness (using 2 kPa, 4.5 kPa and 21 kPa) and independently changing the composition of ECM (COL1 in Fig. 3 versus COL6 in Fig. 7 in the revised manuscript/ old Fig. 6).

We do agree with the Reviewer, and think that 3D models better recapitulate *in vivo* cell responses and partially incorporate the contribution of the tumor microenvironment (TME), particularly the extracellular matrix (ECM), thus we have used these models in our study to investigate global responses to platinum therapy (see Fig. 6 and Fig. 8; Supplementary Fig. 11). Nonetheless, 3D matrices impose physical constrictions that challenge the separation between the effects of ECM remodeling and adhesion, which motivated us to use both 2D and 3D models in our study.

4) The reason that cells responded differently (figure 3G) is not clear, and as such it is not clear how the data should be interpreted.

Response: We thank the Reviewer for the constructive comment and have now performed additional experiments to clarify this issue. The results in the original manuscript indicated that stiffness alters cell-ECM adhesion and intracellular signaling, seen by increased focal adhesions and nuclear localization of YAP/TAZ in stiff COL1-substrate, as expected based on various studies (Figure 3a-d). In addition, both cell proliferation (assessed by Edu incorporation) and cisplatin-induced DNA damage (measured as intensity of phosphorylated histone H2Ax (γ H2Ax) foci) were increased with increasing stiffness (Figure 3e-f). These results were significant only in OVCAR4, but the same trend could be seen in OVCAR8. Coincidentally, increased stiffness resulted in reduced cell death upon cisplatin treatment (Figure 3g), a result which was expected, since stiff microenvironments confer protection to the cancer cells²¹.

However, considering that DNA damage is often used as an indication of upcoming cell death, our results on highest cisplatin-induced DNA damage and lowest cell death in stiff COL1 could seem contradictory. Yet our interpretation was that the stiffness-mediated protection was independent of the extent of DNA damage. To further explore this hypothesis, we inhibited FAK and YAP, resulting in increased apoptosis but not increased DNA damage (see new Fig. 3h-k). These results indicate that the resistance (decreased apoptosis) seen in stiff COL1 is mediated by ECM adhesion signaling, since inhibiting this signaling sensitizes the cells to platinum. DNA damage was not increased coincident with this sensitization, but even strongly decreased, particularly in OVCAR4, indicating that the stiffness-induced resistance is uncoupled from the DNA damage.

In addition, we sought to understand whether the significantly lower γ H2Ax foci intensity in OVCAR8 at high stiffness (Fig. 3f) compared to OVCAR4 could be explained by differences in DNA damage response signaling, as suggested by Reviewer#2 comment 4. Since cells lacking BRCA1 depend on chk1 for the repair of endogenous DNA damage¹⁵ and OVCAR8 have reduced BRCA1 expression due to gene methylation¹⁴, we inhibited chk1/2, downstream of the HA2X phosphorylating ATM/ATR-complexes. As expected, chk1/2 inhibition resulted in sensitization of HGSC cells to cisplatin (Fig. 3l)¹⁶. However, OVCAR8 but not OVCAR4 showed more platinum-induced DNA damage upon chk1/2 inhibition (Fig. 3m), suggesting that chk1/2 mediate protection against DNA damage in OVCAR8.

We have now included these results in Fig. 3 and Supplementary Fig. 6, and described them in Results in pages 10-11, concluding that "... evasion of cisplatin-induced apoptosis via DNA damage response & repair pathways and stiffness-dependent adhesion signaling can occur via distinct, overlapping mechanisms."

5) Much of the data is correlative. As an example, the data on COL6 (Figure 5g-h) is not demonstrated definitively as a protector for chemoresistance. Much of the data and interpretation is based on comparison of 2 cells lines OVCAR4 and OVCAR8 which is not sufficient to draw a conclusion.

Response: In the original manuscript, the protective effect of COL6 seen in OVCAR8 (original Fig. 5g-h) was validated in HGSC patient-derived organoids. Specifically, the cells obtained from patients at relapse-stage (n = 4 r-HGSC; compared to n = 4 pre-chemotherapy HGSC organoids) were more resistant to cisplatin in COL6-containing 3D matrices (original Fig. 7g-h). These results suggested that the protection by COL6 could derive from intrinsic platinum resistance mechanisms that are already active in these cells (OVCAR8 and r-HGSC ascites-derived organoids). Nevertheless, to strengthen our results we have now repeated these experiments using the relatively platinum sensitive OVCAR3 and TYK-nu parental, and the experimentally produced more resistant subline TYK-nu.R^{3,5,6}. Similar to OVCAR8 and r-HGSC, TYK-nu.R cells are protected against cisplatin treatment in the COL6-containing 3D matrices, whereas the more cisplatin sensitive OVCAR3 and TYK-nu are not (see new Fig. 6f-h). Altogether, these results further support our original findings on the COL6-mediated protection against cisplatin in intrinsically chemoresistant cells and have been included in page 16 of the revised manuscript.

References

1. Pearce, O. M. T. *et al.* Deconstruction of a Metastatic Tumor Microenvironment Reveals a Common Matrix Response in Human Cancers. *Cancer Discov.* **8**, 304–319 (2018).
2. Beaufort, C. M. *et al.* Ovarian cancer cell line panel (OCCP): Clinical importance of in vitro morphological subtypes. *PLoS One* **9**, (2014).
3. Yoshiya, N. *et al.* [Isolation of cisplatin-resistant subline from human ovarian cancer cell line and analysis of its cell-biological characteristics]. *Nihon Sanka Fujinka Gakkai Zasshi* **41**, 7–14 (1989).
4. Domcke, S., Sinha, R., Levine, D. A., Sander, C. & Schultz, N. Evaluating cell lines as tumour models by comparison of genomic profiles. *Nat. Commun.* **4**, 2126 (2013).
5. Moyano-Galceran, L. *et al.* Adaptive RSK-EphA2-GPRC5A signaling switch triggers chemotherapy resistance in ovarian cancer. *EMBO Mol. Med.* (2020). doi:10.15252/emmm.201911177
6. Haley, J. *et al.* Functional characterization of a panel of high-grade serous ovarian cancer cell lines as representative experimental models of the disease. *Oncotarget* **7**, 32810–32820 (2016).
7. Gara, S. K. *et al.* Three novel collagen VI chains with high homology to the $\alpha 3$ chain. *J. Biol. Chem.* **283**, 10658–10670 (2008).
8. Sherman-Baust, C. A. *et al.* Remodeling of the extracellular matrix through overexpression of collagen VI contributes to cisplatin resistance in ovarian cancer cells. *Cancer Cell* **3**, 377–386 (2003).
9. Yeung, T. L. *et al.* TGF- β Modulates ovarian cancer invasion by upregulating CAF-Derived versican in the tumor microenvironment. *Cancer Res.* **73**, 5016–5028 (2013).
10. Iyengar, P. *et al.* Adipocyte-derived collagen VI affects early mammary tumor progression in vivo, demonstrating a critical interaction in the tumor/stroma microenvironment. *J. Clin. Invest.* **115**, 1163–76 (2005).
11. Clamp, A. R. *et al.* Weekly dose-dense chemotherapy in first-line epithelial ovarian, fallopian tube, or primary peritoneal carcinoma treatment (ICON8): primary progression free survival analysis results from a GCIg phase 3 randomised controlled trial. *Lancet* **394**, 2084–2095 (2019).
12. Cohen, P. A. *et al.* Pathological chemotherapy response score is prognostic in tubo-ovarian high-grade serous carcinoma: A systematic review and meta-analysis of individual patient data. *Gynecologic Oncology* **154**, 441–448 (2019).

13. Kessenbrock, K., Plaks, V. & Werb, Z. Matrix Metalloproteinases: Regulators of the Tumor Microenvironment. *Cell* **141**, 52–67 (2010).
14. Stordal, B. *et al.* BRCA1/2 mutation analysis in 41 ovarian cell lines reveals only one functionally deleterious BRCA1 mutation. *Mol. Oncol.* **7**, 567–579 (2013).
15. Paculová, H. *et al.* BRCA1 or CDK12 loss sensitizes cells to CHK1 inhibitors. *Tumor Biol.* **39**, 1–11 (2017).
16. Gralewska, P., Gajek, A., Marczak, A. & Rogalska, A. Participation of the ATR/CHK1 pathway in replicative stress targeted therapy of high-grade ovarian cancer. *Journal of Hematology and Oncology* **13**, (2020).
17. Tulla, M. *et al.* Selective Binding of Collagen Subtypes by Integrin α 11, α 21, and α 101 Domains. *J. Biol. Chem.* **276**, 48206–48212 (2001).
18. Barczyk, M., Carracedo, S. & Gullberg, D. Integrins. *Cell and Tissue Research* **339**, 269–280 (2010).
19. Miyazaki, K. *et al.* Collective cancer cell invasion in contact with fibroblasts through integrin- α 5 β 1/fibronectin interaction in collagen matrix. *Cancer Sci.* cas.14664 (2020). doi:10.1111/cas.14664
20. Raab-Westphal, S., Marshall, J. & Goodman, S. Integrins as Therapeutic Targets: Successes and Cancers. *Cancers (Basel)*. **9**, 110 (2017).
21. Henke, E., Nandigama, R. & Ergün, S. Extracellular Matrix in the Tumor Microenvironment and Its Impact on Cancer Therapy. *Frontiers in Molecular Biosciences* **6**, 160 (2020).

Figures for Reviewers

Figure 1 for Reviewers. Genomic analyses of HGSC ascites-derived organoids.

a, Heatmap for ERBB2 and KRAS mutations in high grade serous carcinoma (HGSC) ascites-derived organoids from pre-chemo (p-HGSC) and progressive (r-HGSC) stages. The data that support these findings are deposited in European Genome-phenome Archive under accession code EGAS00001004714.

Figure 2 for Reviewers. Inhibition of integrin β 1 signaling.

a, Representative confocal micrographs of siSCR versus siITG β 1 OVCAR8 grown in 3D COL1 and stained for CD44 (green). Scale bar 50 μ m.

b,c, Charts illustrate cell viability (**b**; ATP-based measurement) and relative apoptosis (**c**; measurement of caspase 3/7 activity) in siSCR versus siITG β 1 OVCAR4, OVCAR8, TYK-nu and TYK-nu.R grown in 3D COL1 and treated with 10 μ M (OVCAR4, TYK-nu and TYK-nu.R) or 20 μ M (OVCAR8) cisplatin for 72 h. Error bars represent standard deviation; n = 3 independent replicates (**b**) or 2 independent experiments with 3 technical replicates each (**c**); two-tailed Student's t-test.

REVIEWER COMMENTS

Reviewer #1 (Remarks to the Author):

Thank you for addressing my questions.

Reviewer #2 (Remarks to the Author):

In this revised manuscript, the authors have addressed many reviewer comments and present new data. However, although the authors have clearly undertaken a significant amount of work, using primary material, which is always challenging, this reviewer is still struggling to identify a clear story or a coherent mechanism beyond the purely descriptive presentation of the results.

Critically, the authors demonstrate clear differences between OVCAR4 and OVCAR8 cells but do not show a clear mechanism to explain the differences. Similarly, Figure 8 shows very interesting differences between pre- and post-chemotherapy cultures grown on COL1 vs COL1+COL6, but without a clear mechanistic insight into a) why and b) what this means for patients and their treatment.

Specific points

1. Figure 2i, j and Supplementary Figure 5, the immune cell infiltration is not 'remarkable'. Similarly, there is no pre-chemotherapy immune IHC and no quantification of these images, so it is hard to make any conclusion here.
2. The link between DNA damage induction (as measured by gammaH2AX) and apoptosis induction is particularly confusing. Firstly, I believe that the authors are present gammaH2AX staining as intensity rather than number of discrete foci. Secondly, it is still not clear why there is a disconnect between DNA damage and apoptosis. The data on OVCAR8 and CHK1/2 inhibition add more confusion. The authors do not make any attempt to look at BRCA1 methylation in their OVCAR8 cells nor to demonstrate whether the cells are or are not able to repair DNA double strand break damage using homology-mediated repair. Increasing gammaH2AX staining can reflect both an increase in DNA damage induction and a reduction in repair such that foci do not resolve. This part of the paper remains unclear but understanding what is going on is critical.
3. Figures 4, 6 and 7 show a series of highly complex assays. However, on repeated reading, this reviewer can only conclude that there are differences between the different cell lines but no mechanistic description (e.g. specific mutation, amplification, deletion, methylation, gene expression in the individual cell lines) that explain clearly and consistently the patterns of behaviour that are presented. The results appear purely descriptive.
4. Figure 5 is also confusing. The authors use the expressions 'disease progression' (line 351) and 'evolution' (line 355) in the pre-chemotherapy samples. However, I think that they mean metastatic sites compared to primary tumour: these samples were all obtained at the same time, so any differences are pre-diagnostic and do not represent any treatment-induced changes. The critical questions that this reviewer wished to know from this figure were a) are there significant differences in expression in any of the genes between primary sites and metastatic sites pre-chemo? And post-chemo?
5. If I understand correctly, Figure 5c suggests that the greater the change in COL6A1 and COL6A2 expression pre- and post-chemotherapy in matched samples, the worse the survival (PFS) when multiple sample types are analysed together. However, 5b appears to show no significant difference

pre- and post-chemotherapy apart from COL6A2 in mesenteric metastases. Thus, it is difficult to know quite what the meaning of this figure is.

Reviewer #3 (Remarks to the Author):

The manuscript takes an important step in helping our understanding of chemo-resistance in ovarian cancer and the role of the tumor microenvironment. Overall the paper is greatly improved but just a few issues, while addressed in the response, were not adequately addressed in the manuscript. . As examples:

-The two cell lines show different responses in several assays, but there is inadequate explanation in the text to explain this. Why were these two lines chosen? The third line as brought in response to reviewers but the logic in the text is not made clear.

-The choice of 2 and 21kPa is not clear. Are these physiologically relevant?

We thank the Reviewers for the continued interest in our work and appreciate the positive yet constructive comments. These motivated us to further conduct new experiments and allowed important improvements to the manuscript as described below.

As requested by the editor, we particularly addressed the concerns of the Reviewers in regards to the rationale behind the use of the different cell lines, the analysis of DNA damage and repair and presentation of the results in Fig.5. In addition, we revised the text to highlight the central message and scientific as well as clinical impact of this study and excluded any claims of remarkable immune cell infiltration.

Reviewer #2 (Remarks to the Author):

In this revised manuscript, the authors have addressed many reviewer comments and present new data. However, although the authors have clearly undertaken a significant amount of work, using primary material, which is always challenging, this reviewer is still struggling to identify a clear story or a coherent mechanism beyond the purely descriptive presentation of the results.

Critically, the authors demonstrate clear differences between OVCAR4 and OVCAR8 cells but do not show a clear mechanism to explain the differences. Similarly, Figure 8 shows very interesting differences between pre- and post-chemotherapy cultures grown on COL1 vs COL1+COL6, but without a clear mechanistic insight into a) why and b) what this means for patients and their treatment.

Specific points

1. Figure 2i, j and Supplementary Figure 5, the immune cell infiltration is not 'remarkable'. Similarly, there is no pre-chemotherapy immune IHC and no quantification of these images, so it is hard to make any conclusion here.

Response: In this revised manuscript, we have removed any claims concerning remarkable immune cell infiltration and the immunohistochemistry (IHC) for specific immune cell types, except CD68 for macrophages and CD45 generally for leukocytes. We have now revised the text to describe this observation as follows (lines 194-196): "These compromised stromal ECM fibers were surrounded by small cells similar to those detected as CD45+ immune cells". In addition, we have removed discussion about possible association between collagen fibers, MMPs and immune cells, and rather just state following "...what are the roles of the other ECM components such as COL5 and COL11 as well as proteolytic enzymes including MMPs also prominent in our matrisome signatures, remains of future interest" (page 21, lines 512-514).

2. The link between DNA damage induction (as measured by gammaH2AX) and apoptosis induction is particularly confusing. Firstly, I believe that the authors are present gammaH2AX staining as intensity rather than number of discrete foci. Secondly, it is still not clear why there is a disconnect between DNA damage and apoptosis. The data on OVCAR8 and CHK1/2 inhibition add more confusion. The authors do not make any attempt to look at BRCA1 methylation in their OVCAR8 cells nor to demonstrate whether the cells are or are not able to repair DNA double strand break damage using homology-

mediated repair. Increasing gammaH2AX staining can reflect both an increase in DNA damage induction and a reduction in repair such that foci do not resolve. This part of the paper remains unclear but understanding what is going on is critical.

Response: We appreciate these important points raised by the Reviewer. For clarity, we have divided our response to address specific issues, as follows:

Issue 1: Firstly, I believe that the authors are present gammaH2AX staining as intensity rather than number of discrete foci.

This is correct, we have measured the intensity of phosphorylated H2Ax (γ H2Ax) per nuclei and not the number of foci. To clarify this issue, we have modified the text in lines 235 and 262, the Y-axis titles of the graphs illustrating γ H2Ax as well as the respective figure legends, and we have also included the nuclear staining (DAPI) in all micrographs to illustrate the total cell content. In addition, we have improved the presentation of the results by showing the intensity of each individual nucleus in superplots (see Fig. 3e,i,k and Supplementary Fig.7c).

Issue 2: Secondly, it is still not clear why there is a disconnect between DNA damage and apoptosis.

To confirm and better understand this disconnection, we have now assessed phosphorylated H2Ax (measured as intensity of γ H2Ax per nuclei) over a 36-hour treatment in time course experiments, together with quantification of cleaved caspase-3/7 and total cell numbers over a 72-hour treatment. These results complement those described in our originally submitted manuscript from a single time point after 32 hours of treatment, which could not fully explain the disconnection between DNA damage and apoptosis.

Our new results show that in all cell lines DNA damage accumulated over 36-h cisplatin treatment on both soft and stiff matrix (Fig. 3e). However, the increase of cisplatin-induced cleaved caspase-3/7 was significantly higher on soft than on stiff COL1 (Fig. 3f), corroborating our previous findings on stiff substrates conferring protection against cisplatin-induced apoptosis. At 36 hours, when the γ H2Ax intensity was significantly higher in cells cultured on stiff than on soft, OVCAR4, OVCAR8, and TYK-nu.R showed higher apoptosis on soft COL1 (OVCAR4 being significant at 39h), indicating that cells enter apoptosis at lower DNA damage on soft than on stiff COL1. However, on stiff substrate, cells progressively showed higher γ H2Ax without entering apoptosis, which explains the difference between soft and stiff conditions observed at the previously shown single time point (32 h). Cleaved caspase 3/7 positivity is significantly higher on stiff than soft substrate at various time points for OVCAR4, OVCAR8, and TYK-nu.R. Altogether, this data indicates that the cells have increased resistance to platinum-mediated, apoptosis-inducing DNA damage on stiff compared to soft COL1. We have added these results in the revised manuscript in pages 10-11. We have also included in supplementary data example videos of the live cell imaging time course of cisplatin treatment for each cell line (phase contrast and cl-casp-3/7; Supplementary Movies 1-4)

Issue 3: The data on OVCAR8 and CHK1/2 inhibition add more confusion.

In the first revision of this manuscript we performed CHK1/2 inhibition to investigate whether the lower γ H2Ax intensity in OVCAR8 compared to OVCAR4 on high stiffness (old Fig. 3f, Supplementary Fig. 7c in this revised manuscript) could be explained by enhanced DNA damage repair in OVCAR8 by other means than HR repair pathway^{2,3}. However, as CHK1/2 not only promote DNA damage repair but also regulate cell cycle arrest⁴, we agree with the Reviewer that these results as such did not answer the question. For this reason, we have removed them in this revised version to clarify the presentation.

Issue 4: The authors do not make any attempt to look at BRCA1 methylation in their OVCAR8 cells nor to demonstrate whether the cells are or are not able to repair DNA double strand break damage using homology-mediated repair.

We have now investigated the homology recombination (HR) efficiency/status of OVCAR4, OVCAR8, TYK-nu and TYK-nu.R by measuring the ability of cells to form RAD51 foci upon cisplatin-induced DNA damage in cyclinA2⁺ cells^{5,6}. In Fig. 3g and Supplementary Fig. 8b-d in this revised manuscript, we show that in both soft and stiff hydrogels, 66-91% of the BRCA-wild type OVCAR4, TYK-nu and TYK-nu.R are RAD51⁺/cyclinA2⁺, whereas the BRCA1-methylated OVCAR8 display only 8-10% RAD51⁺/cyclinA2⁺⁷⁻⁹. As this reduced ability to form RAD51 foci after DNA damage represents a functional readout of defective HR, our results are in agreement with previous studies describing OVCAR8 as HR-deficient and OVCAR4 as HR-proficient^{5,8,9}. We have added this information in the revised manuscript in page 11.

Issue 5: Increasing gammaH2AX staining can reflect both an increase in DNA damage induction and a reduction in repair such that foci do not resolve.

We agree with the Reviewer that the net/effective DNA damage can be the result of DNA damage induction and its repair. When detecting phosphorylated H2AX, which is a surrogate marker of unresolved DNA damage, we are considering both the induced as well as the unsuccessfully repaired DNA damage. We have now revised the manuscript to avoid the association of phosphorylated H2AX to cisplatin-induced DNA damage exclusively by incorporating the terms “effective DNA damage” and “DNA damage accumulation” in results described in pages 10-11.

3. Figures 4, 6 and 7 show a series of highly complex assays. However, on repeated reading, this reviewer can only conclude that there are differences between the different cell lines but no mechanistic description (e.g. specific mutation, amplification, deletion, methylation, gene expression in the individual cell lines) that explain clearly and consistently the patterns of behaviour that are presented. The results appear purely descriptive.

Response: We understand the Reviewers concern. In order to highlight the findings in these figures and to emphasize the relevance of these, despite we do not provide one mechanistic description, we have now thoroughly revised the manuscript.

We demonstrate that cisplatin alters the ECM-cell communication of both the pairs of more cisplatin-sensitive epithelial OVCAR4 / more resistant mesenchymal OVCAR8 and more sensitive TYK-nu / their more resistant subline TYK-nu.R (sharing same genetic background)¹⁰, especially on VTN, FN and COL6, supporting adhesion, migration and

treatment resistance. In 3D COL1-environment COL6 supported treatment escape of those HGSC cells with intrinsic chemoresistance, the OVCAR8 and TYK-nu.R, but not their more sensitive counterparts. Further, cisplatin enhanced the stiff COL6-mediated focal adhesion signaling coincident with increased treatment-resistance in OVCAR8. These results demonstrate that HGSC cells with intrinsic and active chemoresistance machinery have increased apoptosis-evasion ability in stiff COL6-rich environment that acts as a platform for further discovery and potential target for improved treatment response.

4. Figure 5 is also confusing. The authors use the expressions 'disease progression' (line 351) and 'evolution' (line 355) in the pre-chemotherapy samples. However, I think that they mean metastatic sites compared to primary tumour: these samples were all obtained at the same time, so any differences are pre-diagnostic and do not represent any treatment-induced changes. The critical questions that this reviewer wished to know from this figure were a) are there significant differences in expression in any of the genes between primary sites and metastatic sites pre-chemo? And post-chemo?

Response: We agree with the reviewer that the terms "disease progression" and "evolution" should not be used when comparing samples obtained at the same time/surgery (although this is not the case when referring to pre-chemo samples in lines 351-355, since in that section we compared pre- versus post-chemo samples). We have now modified the text to avoid misusing these terms and to clarify the data presentation. However, in our manuscript we have longitudinally collected pre- and post-chemotherapy samples and thus we can discuss the contribution/effect of the treatment in these cases (such as in lines 351-355 of the first revised manuscript).

In our original manuscript, we performed a comprehensive analysis of the matrisome in pre- and post-chemotherapy samples and described gene expression changes between primary and metastatic sites (see Fig. 1c-d and Supplementary Fig.1 a-b for the pre-chemo results and Fig. 2e for post-chemo data). In addition, as requested by the Reviewer, we have now presented the expression data for *COL6*, *FN1* and *VTN* in pre- and post-chemotherapy primary tumors and combined metastatic tissues in a heatmap and analyzed the differences between these two (Fig. 5b in the revised manuscript). In pre- and post-chemotherapy metastatic tissues *COL6A3* (post-chemo only), *COL6A5*, *COL6A6* and *FN1* were upregulated in comparison to the respective primary tumors. We have added these results to the revised manuscript in page 15 and removed the previous analyses of post- versus pre-chemotherapy anatomical location/ sample type.

5. If I understand correctly, Figure 5c suggests that the greater the change in *COL6A1* and *COL6A2* expression pre- and post-chemotherapy in matched samples, the worse the survival (PFS) when multiple sample types are analysed together. However, 5b appears to show no significant difference pre- and post-chemotherapy apart from *COL6A2* in mesenteric metastases. Thus, it is difficult to know quite what the meaning of this figure is.

Response: It is correct that in Fig. 5c of the first revised manuscript we showed the correlation between survival and change in gene expression (post- versus pre-

chemotherapy) in 12 patient- and tissue-matched samples. In contrast, in Fig. 5b we showed the change in expression (post- versus pre-chemotherapy) in bulk patient samples from matching anatomical locations.

In order to simplify Fig. 5c presentation and the corresponding text, we now show the chemo-induced gene expression changes in metastatic tumors from individual patients and the association to treatment response: progression free survival and platinum-free interval. For those patients with more than one metastatic tissue sample we have now merged the different metastatic sites as well as removed the primary tumors from the analysis. Each dot represents the change in expression in post-chemotherapy omental, peritoneal or mesenteric metastatic tissues (alone or combined) in comparison to the corresponding pre-chemotherapy tissue sites in each individual patient. We have modified the text in page 15 accordingly.

Reviewer #3 (Remarks to the Author):

The manuscript takes an important step in helping our understanding of chemo-resistance in ovarian cancer and the role of the tumor microenvironment. Overall the paper is greatly improved but just a few issues, while addressed in the response, were not adequately addressed in the manuscript. As examples:

1. The two cell lines show different responses in several assays, but there is inadequate explanation in the text to explain this. Why were these two lines chosen? The third line as brought in response to reviewers but the logic in the text is not made clear.

Response: We thank the Reviewer for the continued interest in our work and for the constructive comments. We have now thoroughly revised the text and described the rationale for the selection and use of the different cell lines (see lines 208-212 and 297-299).

All the cells used in this manuscript recapitulate the mutational landscape of HGSC⁷. However, as HGSC is a very heterogeneous disease, we selected cells with different phenotypes in terms of their ability to invade, migrate, and proliferate as well as with varied chemoresistance and epithelial-mesenchymal status to recapitulate the diversity seen in HGSC patient-derived cells¹¹⁻¹³. In this study/manuscript, we have used the relatively platinum-sensitive, epithelial ($CDH1^+$, $CDH2^{low}$) OVCAR4, the more resistant and mesenchymal ($CDH1^-$, $CDH2^+$) OVCAR8, and the platinum-sensitive, mesenchymal ($CDH1^-$ & $CDH2^{low}$) TYK-nu. In order to model the intrinsic chemoresistance of post-chemo/relapse cells, we included a more platinum-resistant subline of TYK-nu, the TYK-nu.R, generated by repeated cisplatin exposure¹⁰. In our first revised manuscript, a fifth cell line, OVCAR3, was also included in selected experiments in order to have another more epithelial ($CDH1^+$, $CDH2^{low}$) sensitive HGSC cell model (Fig. 5f, 6f and Supplementary Fig. 10i-j)¹⁴.

2. The choice of 2 and 21kPa is not clear. Are these physiologically relevant?

Response: The specific selection of 2 kPa and 21 kPa was both for technical reasons and even more importantly to mimic the physiological stiffness of chemo-naïve human omental metastases^{15,16}. Previously, Pearce O *et al* reported a stiffness variation from 0.40 kPa to 33.13 kPa in 32 HGSC omental metastatic biopsies with one extremely stiff 61.90 kPa sample. Based on this study, the average of the soft tumor is around 2 kPa and around 20 kPa in stiff tissues (excluding the eccentric 61.90 kPa). To clarify the relevance of the selected stiffness we have now modified the manuscript as follows (lines 204-208): To examine first the signaling response of HGSC cells to in vivo-like, physiological low and high stiffness range of HGSC omental metastasis tissues (0.40-33.13 kPa), we used the soft (2kPa) and stiff (21 kPa) polyacrylamide hydrogels functionalized for cell adhesion with covalently-bound COL1.

References

1. Heylmann, D. & Kaina, B. The γ H2AX DNA damage assay from a drop of blood. *Sci. Rep.* **6**, 1–9 (2016).
2. Gralewska, P., Gajek, A., Marczak, A. & Rogalska, A. Participation of the ATR/CHK1 pathway in replicative stress targeted therapy of high-grade ovarian cancer. *Journal of Hematology and Oncology* vol. 13 (2020).
3. Paculová, H. *et al*. BRCA1 or CDK12 loss sensitizes cells to CHK1 inhibitors. *Tumor Biol.* **39**, 1–11 (2017).
4. Patil, M., Pabla, N. & Dong, Z. Checkpoint kinase 1 in DNA damage response and cell cycle regulation. *Cellular and Molecular Life Sciences* vol. 70 4009–4021 (2013).
5. Tumiati, M. *et al*. A functional homologous recombination assay predicts primary chemotherapy response and long-term survival in ovarian cancer patients. *Clin. Cancer Res.* **24**, 4482–4493 (2018).
6. San Filippo, J., Sung, P. & Klein, H. Mechanism of Eukaryotic Homologous Recombination. *Annu. Rev. Biochem.* **77**, 229–257 (2008).
7. Domcke, S., Sinha, R., Levine, D. A., Sander, C. & Schultz, N. Evaluating cell lines as tumour models by comparison of genomic profiles. *Nat. Commun.* **4**, 2126 (2013).
8. Stordal, B. *et al*. BRCA1/2 mutation analysis in 41 ovarian cell lines reveals only one functionally deleterious BRCA1 mutation. *Mol. Oncol.* **7**, 567–579 (2013).
9. Pulliam, N. *et al*. An effective epigenetic-PARP inhibitor combination therapy for breast and ovarian cancers independent of BRCA mutations. *Clin. Cancer Res.* **24**, 3163–3175 (2018).
10. Yoshiya, N. *et al*. [Isolation of cisplatin-resistant subline from human ovarian cancer cell line and analysis of its cell-biological characteristics]. *Nihon Sanka Fujinka Gakkai Zasshi* **41**, 7–14 (1989).
11. Winterhoff, B. J. *et al*. Single cell sequencing reveals heterogeneity within ovarian cancer epithelium and cancer associated stromal cells. *Gynecol. Oncol.* **144**, 598–606 (2017).
12. Verhaak, R. G. W. *et al*. Prognostically relevant gene signatures of high-grade serous ovarian carcinoma. *J. Clin. Invest.* **123**, 517–525 (2013).
13. Chirshv, E. *et al*. Epithelial/mesenchymal heterogeneity of high-grade serous ovarian carcinoma samples correlates with miRNA let-7 levels and predicts tumor growth and metastasis. *Mol. Oncol.* **14**, 2796–2813 (2020).
14. Moyano-Galceran, L. *et al*. Adaptive RSK-EphA2-GPRC5A signaling switch triggers chemotherapy resistance in ovarian cancer. *EMBO Mol. Med.* (2020)

doi:10.15252/emmm.201911177.

15. Pearce, O. M. T. *et al.* Deconstruction of a Metastatic Tumor Microenvironment Reveals a Common Matrix Response in Human Cancers. *Cancer Discov.* **8**, 304–319 (2018).
16. Trappmann, B. *et al.* Extracellular-matrix tethering regulates stem-cell fate. *Nat. Mater.* **11**, 642–649 (2012).

REVIEWERS' COMMENTS

Reviewer #2 (Remarks to the Author):

The authors have made extensive efforts to modify the manuscript in response to reviewer comments. The manuscript is huge and complex and requires a great deal of reader concentration.

Nonetheless, there is an important message about ECM stiffness and signalling in HGSC and its effects on response to platinum. Thus, I am happy with the changes that the authors have made and feel that they have addressed my previous comments

Reviewer #2 (Remarks to the Author):

The authors have made extensive efforts to modify the manuscript in response to reviewer comments. The manuscript is huge and complex and requires a great deal of reader concentration.

Nonetheless, there is an important message about ECM stiffness and signalling in HGSC and its effects on response to platinum. Thus, I am happy with the changes that the authors have made and feel that they have addressed my previous comments

Response: We thank the Reviewer for the continued interest in our work and appreciate the positive comments. In order to clarify the reporting of our findings in this manuscript we have further revised the Discussion, especially lines 544-558. To simplify the presentation, we have also moved the old Fig 7a,b to Supplementary Fig. 14.